# From Perception to Planning: Evolving Ego-Centric Task-Oriented Spatiotemporal Reasoning via Curriculum Learning

**Xiaoda Yang** [* 1]  **Yuxiang Liu** [* 2]  **Shenzhou Gao** [3]  **Can Wang** [3]  **Jingyang Xue** [3]  **Lixin Yang** [4]  **Yao Mu** [4]  **Tao Jin** [1]
**Zhimeng Zhang** [1]  **Shuicheng Yan** [5]  **Zhou Zhao** [1]

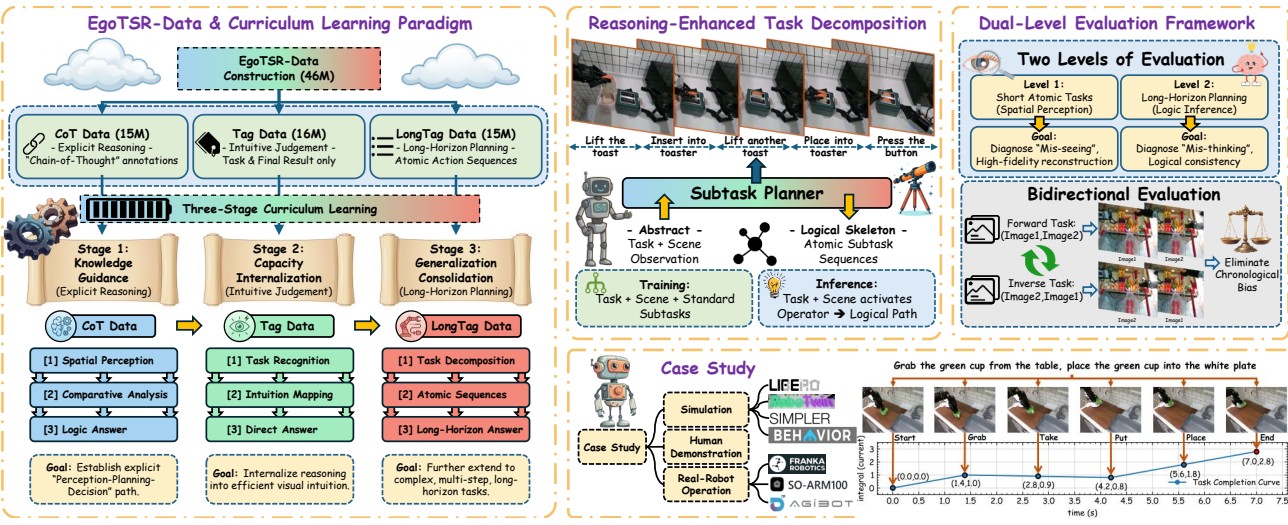

*Figure 1.* **Overview of the EgoTSR Framework.** This figure illustrates the evolving ego-centric task-oriented spatiotemporal reasoning process via curriculum learning. **Left:** The EgoTSR-Data of 46 million samples and the three-stage curriculum learning paradigm, transitioning from knowledge guidance (CoT Data) to capacity internalization (Tag Data) and generalization consolidation (LongTag Data). **Top-Middle:** Reasoning-enhanced task decomposition, where a Subtask Planner maps abstract task and scene observations into a logical skeleton of atomic subtask sequences. **Top-Right:** A dual-level evaluation framework focusing on spatial perception and logical inference, incorporating bidirectional evaluation to eliminate chronological bias. **Bottom-Right:** Extensive case studies spanning human demonstrations, simulation environments, and real-robot platforms validate that our EgoTSR has successfully internalized a generalized "Perception-Planning-Decision" cognitive pathway, demonstrating exceptional stability and adaptability in dynamic, open-world scenarios.

## Abstract

Modern vision-language models achieve strong performance in static perception, but remain limited in the complex spatiotemporal reasoning required for embodied, egocentric tasks. A major source of failure is their reliance on temporal priors learned from passive video data, which often leads to spatiotemporal hallucinations and poor generalization in dynamic environments. To address this, we present EgoTSR, a curriculum-based framework for learning task-oriented spatiotemporal reasoning. EgoTSR is built on the premise that embodied reasoning should evolve from explicit spatial understanding to internalized task-state assessment and finally to long-horizon planning. To support this paradigm, we construct EgoTSR-Data, a large-scale dataset comprising 46 million samples organized into three stages: Chain-of-Thought supervision for explicit reasoning, weakly supervised tagging for internalizing reasoning, and long-horizon sequences composed of orthogonal atomic subtasks. We further introduce a Reasoning-Enhanced Task Decomposition mechanism that explicitly models causal dependencies between actions and objects, enabling controllable planning from high-level task descriptions. To rigorously validate our approach,

*Equal contribution  [1]Zhejiang University, School of Computer Science, Zhejiang, China  [2]Tianjin University, School of Computer Science, Tianjin, China  [3]Qingdao University, Shandong, China  [4]Shanghai Jiao Tong University, Shanghai, China  [5]National University of Singapore, Singapore. Correspondence to: Zhou Zhao <zhaozhou@zju.edu.cn>.

*Proceedings of the 43rd International Conference on Machine Learning*, Seoul, South Korea. PMLR 306, 2026. Copyright 2026 by the author(s).

we establish a Dual-Level Evaluation Framework that covers both atomic spatial perception and logical planning. Extensive experiments demonstrate that EgoTSR effectively eliminates chronological biases, achieving 92.4% accuracy on long-horizon logical reasoning tasks while maintaining 88.2% fine-grained perceptual precision, significantly outperforming existing open-source and closed-source state-of-the-art models. The code is available at https://github.com/Collab-Gen/EgoTSR.

## 1. Introduction

Moving towards General-purpose Embodied AI (Hu et al., 2023; Black et al.; Zitkovich et al., 2023) requires a fundamental shift: robots must move beyond isolated manipulation skills (O'Neill et al., 2024; Batra et al., 2020) and reliably accomplish long-horizon tasks in unstructured real-world environments (Fu et al., 2024; Ahn et al., 2022; Huang et al., 2022; Wu et al., 2023; Yang et al., 2025a)—such as "tidying a cluttered desk" or "preparing a meal". In ego-centric settings, such tasks inevitably involve complex and dynamic interactions, where action failures, partial progress, and state regressions are common, making task-oriented spatiotemporal reasoning essential for evaluating whether actions genuinely advance the task toward completion (Sener et al., 2022; Grauman et al., 2022; 2024; Li et al., 2025a). However, a critical vulnerability emerges when deploying existing Vision-Language Models (VLMs) in these settings. Trained largely through passive observation on datasets with fixed causal orderings and limited counterfactual variation (Kung et al., 2025), these models often become overconfident in inferred temporal progression, implicitly assuming that actions unfolding over time correspond to task advancement (Upadhyay et al., 2025). This misplaced confidence leads to severe spatiotemporal hallucinations when comparing multiple frames or interpreting dynamic interactions (Liu et al., 2024; Xu et al., 2024). This limitation highlights that true embodied intelligence requires more than recognizing spatial facts or tracking temporal sequences; models must reason about what a spatial state implies for the task, assess its relevance to the goal, and determine whether the system is genuinely closer to completion (Lu et al., 2025), thereby providing a logical basis for how to act in complex, long-horizon environments.

To address these challenges, we construct **EgoTSR-Data** of 46 million samples and adopt a **Curriculum Learning Paradigm** inspired by human cognitive development (Figure 1, Left). Training initiates with 15M Chain-of-Thought samples to establish an explicit "Perception-Planning-Decision" pathway. Subsequently, we transition to 16M Tag samples and use weak supervision to internalize explicit reasoning into intuitive judgments. Finally, we

utilize 15M Long-Horizon samples of orthogonal atomic operations to consolidate generalization. This "easy-to-hard, explicit-to-internalized" strategy effectively rectifies spatiotemporal hallucinations, empowering the model with reliable long-range planning capabilities.

To bridge high-level semantics and low-level execution, we propose a **Reasoning-Enhanced Task Decomposition** mechanism (Figure 1, Top-Middle). This mechanism decomposes generic task descriptions into sequences of orthogonal atomic steps. By explicitly modeling causal dependencies, it transforms implicit planning into a controllable logical path, significantly enhancing accuracy and robustness in long-horizon task execution.

Furthermore, we establish a **Dual-Level Evaluation Framework** (Figure 1, Top-Right) spanning human demonstrations, simulations, and real-robot experiments. This benchmark evaluates both fine-grained atomic execution and long-horizon spatiotemporal planning, providing a new standard for quantifying logical reasoning in Embodied AI.

To validate the effectiveness of the EgoTSR framework, we conduct extensive experiments. Our results demonstrate that EgoTSR significantly outperforms existing SOTA models, achieving 92.4% accuracy on long-range logical reasoning tasks while maintaining 88.2% fine-grained perceptual precision, resolving the trade-off between atomic perception and global planning. Through ablation studies, we verify that the Curriculum Learning Paradigm is indispensable for eliminating chronological bias, while the Reasoning-Enhanced Task Decomposition mechanism serves as the decisive factor for maximizing accuracy in complex logical dependencies. Furthermore, we analyze the training trajectory and illustrate the model's oscillatory convergence toward robust bidirectional reasoning. Finally, (Figure 1, Bottom-Right) we conduct extensive case studies across human demonstrations, simulation environments (e.g., LIBERO (Liu et al., 2023a), SIMPLER (Li et al., 2024)), and real-robot platforms (e.g., Franka, Agibot, So-100). These qualitative evaluations confirm that EgoTSR has successfully internalized a generalized "Perception-Planning-Decision" cognitive pathway, demonstrating exceptional stability and adaptability in dynamic, open-world scenarios.

In summary, our main contributions are as follows:

- We introduce **EgoTSR-Data** (46M samples) and the **Curriculum Learning Paradigm**, transitioning from CoT reasoning to internalized intuitive judgment and finally towards long-horizon reasoning.

- We propose a **Reasoning-Enhanced Task Decomposition** mechanism that autonomously decomposes generic instructions into several orthogonal atomic steps, bridging the large gap between the high-level semantics of tasks and the low-level execution of robots.

- We establish a **Dual-Level Evaluation Framework** to concurrently assess both atomic perception and long-horizon planning for spatial-temporal reasoning tasks.

## 2. Related Work

**Multi-modal CoT and Curriculum Learning.** Chain-of-Thought (CoT) prompting enhances LLMs by eliciting logical steps (Wei et al., 2022; Kojima et al., 2022), yet in multi-modal domains, it is often limited to inference-time prompting (Zhang et al., 2023; Wang et al., 2025), disconnecting visual perception from logical deduction (Li et al., 2023b; Guan et al., 2024). Recent works also explore structured visual guidance, such as mask-supervised prompt generation in Diff-Prompt (Yan et al., 2025) and executable reasoning plans in Unified Thinker (Zhou et al., 2026). Recent surveys on fine-grained multimodal large language models also emphasize the importance of granular visual perception for multimodal reasoning (Peng et al., 2025). Closely related to our setting, Progressive Training (Yang et al., 2026a) studies curriculum-based VLM training for mitigating spatio-temporal hallucinations in embodied reasoning. To address the remaining gap in dynamic first-person scenarios, we propose a progressive curriculum learning paradigm that simulates human cognition (Bengio et al., 2009; Wang et al., 2021). Moving beyond end-to-end supervision that risks "catastrophic forgetting" (Kirkpatrick et al., 2017; Kemker et al., 2018; Li & Hoiem, 2017; Lopez-Paz & Ranzato, 2017), our framework evolves from explicit spatial reasoning (CoT) to intuitive judgment (Tag) and long-horizon planning (LongTag). This "explicit-to-implicit" evolution ensures sophisticated abstract reasoning while preserving high-fidelity spatial awareness.

**Spatiotemporal Reasoning and Long-Horizon Planning.** While VLMs excel in static scenes (Floridi & Chiriatti, 2020; Team et al., 2023; Zhang et al., 2024), deploying them in dynamic, task-oriented environments remains challenging due to intertwined vulnerabilities: chronological bias in reasoning (Patraucean et al., 2023) and logical drift in planning (Valmeekam et al., 2022). HVD (Xie et al., 2026) further explores human vision-driven video representation via key-frame selection and salient entity modeling, but remains focused on retrieval-oriented cross-modal alignment rather than causal task reasoning. Recent multimodal systems also investigate synchronization, data augmentation, camera-controllable generation, controllable retrieval, and alignment evaluation (Yang et al., 2024b;a; 2025c;b; 2026b), but they do not directly address egocentric causal reasoning over long-horizon task progress. Existing benchmarks (Johnson et al., 2017; Hudson & Manning, 2019; Hong et al., 2023) often neglect complex causal logic, while traditional black-box methods (Kalashnikov et al., 2018; Andrychowicz et al., 2020) or simple state classifiers (Li et al., 2025b;

Zitkovich et al., 2023) lack the granularity to distinguish structurally similar stages (Ma et al., 2023; Kwon et al., 2023). Unlike approaches relying on simple data augmentation (Zheng et al., 2025), our framework fundamentally decouples reasoning from sequential order. By explicitly modeling causal dependencies through a "logical skeleton" (Ji et al., 2025) and enforcing bidirectional logical contrast, we transform implicit planning into a verifiable deduction process, effectively mitigating spatiotemporal hallucinations and ensuring robustness under extended temporal spans.

## 3. Methods

We present EgoTSR, a framework for evolving egocentric spatiotemporal reasoning through three pillars: (subsection 3.1) a Curriculum Learning Paradigm that mimics human cognitive development via three-stage data subsets; (subsection 3.2) a Reasoning-Enhanced Task Decomposition mechanism bridging high-level goals with atomic actions; and (subsection 3.3) a Dual-Level Evaluation Framework that assesses both spatial perception and logical consistency while mitigating chronological biases.

We formulate the problem as learning a policy $\pi_\theta$ that maps visual observations and task instructions to spatiotemporal judgments and planning sequences (Liu et al., 2023b). Let $\mathcal{V} = \{I_1, I_2, \ldots, I_T\}$ denote an ego-centric video sequence, where $I_t \in \mathbb{R}^{H \times W \times 3}$. Given a task instruction $\mathcal{L}_{task}$ and a pair of frames $(I_a, I_b)$ sampled from $\mathcal{V}$, the model must determine the state completion relationship $y \in \{I_a, I_b\}$ and, in complex scenarios, generate a logical plan $\mathcal{S}$.

### 3.1. Large-Scale Dataset and Curriculum Paradigm

#### 3.1.1. DATASET FORMULATION

We construct **EgoTSR-Data**, denoted as $\mathcal{D}$, which is composed of three subsets corresponding to progressive difficulty levels: $\mathcal{D} = \mathcal{D}_{CoT} \cup \mathcal{D}_{Tag} \cup \mathcal{D}_{Long}$ (as shown in Figure 2, Middle). Detailed construction pipeline is shown in Appendix A and sample data are shown in Figure 7. Each sample is a tuple $x_i = (I_a, I_b, \mathcal{L}_{task}, y_{gt}, \mathcal{C})$, where $y_{gt}$ is the ground truth label indicating the frame closer to completion, and $\mathcal{C}$ represents auxiliary context (e.g., reasoning paths or subtask sequences) depending on the subset.

- **CoT (Chain-of-Thought) Data** ($\mathcal{D}_{CoT}$)**:** Focuses on atomic operations. $\mathcal{C}$ includes fine-grained spatial descriptions and implicit logical steps (Wei et al., 2022).

- **Tag Data** ($\mathcal{D}_{Tag}$)**:** Focuses on intuitive judgment. $\mathcal{C} = \emptyset$, forcing the model to map visual inputs directly to $y_{gt}$ without intermediate text.

- **LongTag Data** ($\mathcal{D}_{Long}$)**:** Focuses on long-horizon planning. $\mathcal{C}$ contains the sequence of orthogonal atomic

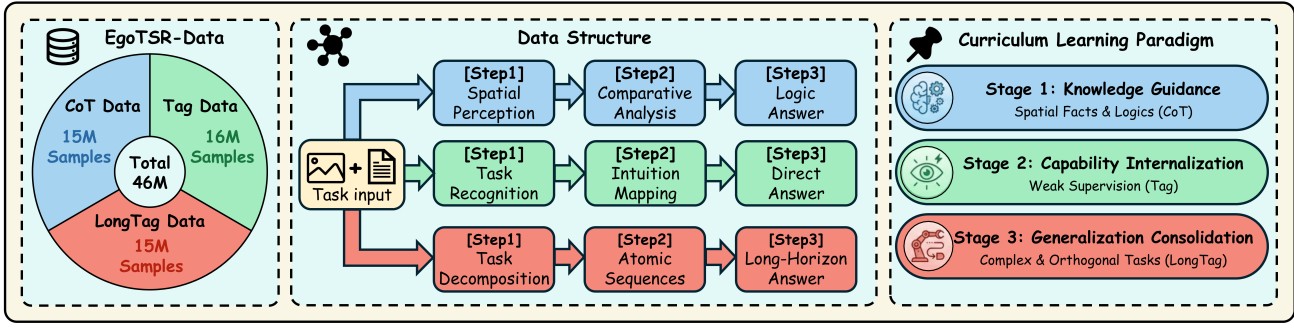

*Figure 2.* Overview of the EgoTSR-Data composition and the three-stage curriculum learning paradigm. Here shows the data structure and the framework evolving from explicit spatial reasoning to internalized intuitive judgment, and finally to complex long-horizon task planning. Different colors correspond to the three types of EgoTSR-data and three stages in the Curriculum Learning Paradigm: CoT (Blue), Tag (Green) and LongTag (Red). For detailed data structure and examples, please refer to **Figure 7**.

subtasks $\mathcal{S} = \{s_1, \ldots, s_k\}$.

$$C = \begin{cases} \mathcal{C}_{cot}, & \text{if } x_i \in \mathcal{D}_{CoT} \\ \emptyset, & \text{if } x_i \in \mathcal{D}_{Tag} \\ \mathcal{S}, & \text{if } x_i \in \mathcal{D}_{Long} \end{cases} \quad (1)$$

### 3.1.2. CURRICULUM LEARNING PARADIGM

We define the training process as a three-stage optimization problem that minimizes the loss function $\mathcal{L}_{\text{stage}}(\theta)$.

**Stage 1: Knowledge Guidance (Explicit Reasoning).** In this stage, the model is trained on CoT Data, where each sample is formulated as:

$$x_{\text{CoT}} = (I_a, I_b, \mathcal{L}_{\text{task}}, y_{\text{gt}}, \mathcal{C}_{\text{CoT}}) \quad (2)$$

We optimize $\theta$ to maximize the likelihood of generating the reasoning path $\mathcal{C}_{\text{cot}}$ followed by the conclusion $y_{\text{gt}}$. The objective function is:

$$\mathcal{L}_{\text{S1}} = -\mathbb{E}_{x \sim \mathcal{D}_{\text{CoT}}} \left[ \log P_\theta(\mathcal{C}_{\text{CoT}}, y_{\text{gt}} | I_a, I_b, \mathcal{L}_{\text{task}}) \right] \quad (3)$$

By mandating fine-grained spatial descriptions and step-by-step logic, we force the model to establish strong correlations between visual pixels and abstract instructions. This explicit logical training not only implicitly maps out paths for complex tasks but also fundamentally constructs a transparent cognitive architecture comprising "environment perception, logical planning, and action decision".

**Stage 2: Capability Internalization (Intuitive Judgment).** In this stage, the model transitions to Tag Data for capability internalization, where each training sample is defined as:

$$x_{\text{Tag}} = (I_a, I_b, \mathcal{L}_{\text{task}}, y_{\text{gt}}) \quad (4)$$

Transitioning to $\mathcal{D}_{\text{Tag}}$, we suppress the explicit reasoning output. The objective simplifies to weak supervision:

$$\mathcal{L}_{\text{S2}} = -\mathbb{E}_{x \sim \mathcal{D}_{\text{Tag}}} \left[ \log P_\theta(y_{\text{gt}} | I_a, I_b, \mathcal{L}_{\text{task}}) \right] \quad (5)$$

Once fundamental logical capabilities are established, training transitions to the Tag data stage, centered on task states. Utilizing weak supervision, we guide the model to reduce its reliance on lengthy textual descriptions and instead focus on the acute capture of critical task nodes. This stage aims to "internalize" the explicit reasoning from the previous stage into an efficient visual intuition. The model achieves a leap in cognitive efficiency, accurately identifying the physical span between the current state and the task goal without recounting every spatial detail.

**Stage 3: Generalization Consolidation (Long-Horizon Planning).** In the final stage, we employ LongTag Data to consolidate generalization across extended temporal spans, formulated as:

$$x_{\text{LongTag}} = (I_a, I_b, \mathcal{L}_{\text{task}}, y_{\text{gt}}, \mathcal{C}_{\text{LongTag}}) \quad (6)$$

For long-horizon tasks, we incorporate the logical skeleton $\mathcal{S}$ (derived in subsection 3.2) as a condition. The loss ensures consistent reasoning across multi-step operations:

$$\mathcal{L}_{\text{S3}} = -\mathbb{E}_{x \sim \mathcal{D}_{\text{LongTag}}} \left[ \log P_\theta(y_{\text{gt}} | I_a, I_b, \mathcal{L}_{\text{task}}, \mathcal{S}) \right] \quad (7)$$

Finally, the model is challenged with LongTag scenarios involving extended temporal spans and complex environmental perturbations. The model must maintain consistent reasoning logic across multi-step operations. This rigorous training in large-scale spatiotemporal contexts consolidates the model's global control over long-horizon tasks. It not only tests the boundaries of generalization but also ensures adaptability to unpredictable visual changes in real-world dynamic scenes.

This "easy-to-hard, explicit-to-internalized" strategy ($\mathcal{L}_{\text{S1}} \rightarrow \mathcal{L}_{\text{S2}} \rightarrow \mathcal{L}_{\text{S3}}$) effectively rectifies common spatiotemporal hallucinations in VLMs (Li et al., 2023b; Guan et al., 2024) at the algorithmic level. More importantly, it endows the agent with reliable long-range planning and execution capabilities at the cognitive level, ensuring high logical robustness and spatial adaptability when facing high-complexity embodied tasks.

## 3.2. Reasoning-Enhanced Task Decomposition

### 3.2.1. DECOMPOSITION FORMULATION

To bridge the gap between abstract instructions and atomic execution in long-horizon spatiotemporal reasoning tasks (Ahn et al., 2022; Yao et al., 2022), we define a task-decomposition mechanism. Formally, $\Phi$ is the decomposition mapping function performed by our **Subtask Planner**:

$$\mathcal{S} = \{s_1, s_2, \ldots, s_K\} = \Phi(\mathcal{I}_0, \mathcal{L}_{\text{task}}) \qquad (8)$$

where $\mathcal{I}_0$ is the initial scene observation and each $s_k$ represents an orthogonal atomic subtask.

Given an abstract task name coupled with visual observations of the scene, the mechanism deconstructs the goal into a series of atomic subtasks that are orthogonal in the functional scope (Zhou et al., 2022). For instance, the long-horizon task "Open the fridge to get food" is decomposed to "Open the refrigerator door with the left arm", "Pick up the cola from the fridge with the left arm", "Place the cola held in the left arm on the table" and "Push the refrigerator door with the right arm" through our mechanism.

### 3.2.2. SUBTASK PLANNER OPTIMIZATION

The Subtask Planner acts as a structured prior. Unlike end-to-end approaches that may suffer from logical drift (Wang et al., 2023; Valmeekam et al., 2022; Shridhar et al., 2020), we explicitly train the planner to model causal dependencies. The optimization objective for the planner is defined as standard next-token prediction over the subtask sequence:

$$\mathcal{L}_{\text{Plan}} = -\sum_{k=1}^{K} \log P(s_k | \mathcal{I}_0, \mathcal{L}_{\text{task}}, s_{<k}) \qquad (9)$$

**Training-Inference Alignment.** To ensure stability in practical applications, we emphasize the establishment of alignment in training and inference phases:

- **Training:** The model receives the full tuple $(\mathcal{I}_0, \mathcal{L}_{\text{task}}, \mathcal{S}_{\text{gt}})$ to learn the mapping $\Phi$. Through strong supervision, it establishes an explicit mapping between task names and physical logic.

- **Inference:** The model autonomously generates $\hat{\mathcal{S}} = \Phi(\mathcal{I}_0, \mathcal{L}_{\text{task}})$. This design ensures that, despite differing input modes, the logical paths remain highly aligned, avoiding the behavioral divergence typical of end-to-end models in long sequences.

In the overall evolution of our architecture, this operator bridges the gap between high-level semantics and physical execution. By coupling micro-discriminative phases (Stages 1 & 2) with macro-guidance (Stage 3), it transforms implicit planning into explicit operational guidelines.

This logical guidance enhances global control and enables self-calibration through subtask feedback, achieving robust long-horizon task execution in complex environments.

## 3.3. Dual-Level Evaluation Framework

### 3.3.1. TASK DIMENSIONALITY

To rigorously decouple the spatial perception from temporal planning, we formalize our evaluation architecture $\mathcal{E}$ into two hierarchical subspaces (Li et al., 2023a): $\mathcal{T}_{\text{short}}$ and $\mathcal{T}_{\text{long}}$. Detailed prompt templates are shown in Appendix B. This enables a qualitative factorization of the total error into spatial and temporal components:

**Level 1: Atomic Perception** ($\mathcal{T}_{\text{short}}$). Errors in $\mathcal{T}_{\text{short}}$ indicate a failure in processing environmental feedback, termed *"mis-seeing"* (Guan et al., 2024). The short-range perception test evaluates the model's sensitivity to fine-grained physical changes. Let a video segment be denoted as $V$, and two sampled frames as $I_t$ and $I_{t+\Delta t}$. The objective is to determine the state progression based solely on visual feature comparison $\mathcal{F}_{\text{vis}}(I_t, I_{t+\Delta t})$. To ensure rigor, we define a **Multi-Granularity** sampling scheme where the frame interval $\Delta t$ ($\times 10$) serves as the difficulty hyperparameter. The sampling space $\Omega_{\text{short}}$ is defined as:

$$\Omega_{\text{short}} = \{(I_t, I_{t+\Delta t}) \mid \Delta t \in \{5, 6, \ldots, 11, 12+\}(\times 10)\} \qquad (10)$$

In this setting, the model functions as a "Spatial Perception Expert," minimizing the perceptual reconstruction error without the aid of long-term logical context.

**Level 2: Planning Consistency** ($\mathcal{T}_{\text{long}}$). Errors in $\mathcal{T}_{\text{long}}$ (given accurate perception) indicate a failure in long-term decision making, termed *"mis-thinking"* (Guan et al., 2024). The long-range consistency test evaluates logical coherence across a sequence of subtasks $\mathcal{S} = \{s_1, s_2, \ldots, s_K\}$. Given two frames $I_a, I_b$ and the generated logical skeleton $\mathcal{S}$, let $\phi(I) \in \{1, \ldots, K\}$ be a mapping function that returns the subtask index of a given frame.

To prevent overfitting to fixed-length sequences, we introduce a **Triple-Level** sampling window based on the logical distance $d = \phi(I_b) - \phi(I_a)$ (assuming $t_b > t_a$):

$$\text{Win}(I_a, I_b) = \begin{cases} \text{INTRA-TASK}, & \text{if } \phi(I_a) = \phi(I_b) \\ \text{INTER-TASK}, & \text{if } \phi(I_b) = \phi(I_a) + 1 \\ \text{MULTI-TASK}, & \text{if } |\phi(I_b) - \phi(I_a)| \geq 2 \end{cases} \qquad (11)$$

By imposing constraints via explicit orthogonal atomic operations within $\mathcal{S}$, this level tests the model's ability to utilize the logical skeleton to identify the global state completion.

*Table 1.* Comprehensive performance on the **Dual-Level Evaluation Framework**. The left part corresponds to the **Short** level, while the right part details the **Long** level. The best/second-best results are highlighted in **bold**/underlined (excluding Human Perception).

| | MODEL | LEVEL 1: SHORT (ACC %) | | | | | | | | | LEVEL 2: LONG (ACC %) | | | | | | |
| | | FRAME INTERVAL (×10) | | | | | | | | AVG. | FORWARD | | | INVERSE | | | AVG. |
| | | 5 | 6 | 7 | 8 | 9 | 10 | 11 | ≥12 | | S | M | L | S | M | L | |
|---|---|---|---|---|---|---|---|---|---|---|---|---|---|---|---|---|---|
| | HUMAN PERCEPTION | 97.3 | 97.3 | 98.2 | 99.1 | 99.1 | 100.0 | 100.0 | 100.0 | 98.9 | 94.3 | 95.5 | 95.6 | 93.9 | 96.1 | 95.9 | 95.2 |
| CLOSE-SOURCE | GPT-O4MINI | 70.0 | 70.3 | 72.8 | 67.7 | 60.0 | 67.5 | 67.7 | 70.3 | 68.3 | 64.9 | 69.0 | 85.7 | 34.6 | 30.3 | 42.5 | 55.0 |
| | GPT-4O | 71.1 | 71.7 | 76.7 | 77.8 | 79.2 | 79.2 | 79.2 | 79.2 | 76.8 | 56.6 | 61.2 | 69.7 | 57.3 | 51.5 | 54.6 | 58.2 |
| | GPT-O3 | 67.2 | 61.4 | 59.6 | 61.6 | 59.6 | 64.0 | 62.0 | 66.6 | 62.8 | 71.9 | 64.7 | 73.1 | 69.6 | 66.7 | 68.6 | 69.0 |
| | GEMINI-2.5-PRO | 71.3 | 75.0 | 72.8 | 75.0 | 77.2 | 80.9 | 89.7 | **100.0** | 80.2 | 74.1 | 78.1 | 64.0 | 63.2 | 42.1 | 57.5 | 62.0 |
| | DOUBAO-1.5-PRO | 64.0 | 76.0 | 56.0 | 46.0 | 56.9 | 86.0 | **90.0** | 96.0 | 71.4 | 68.4 | 58.3 | 58.8 | 45.2 | 48.2 | 29.4 | 52.0 |
| | SEED-1.6 | 66.2 | 66.8 | 70.6 | 73.3 | 83.2 | 82.9 | 86.8 | 98.3 | 78.5 | 46.5 | 46.3 | 28.6 | 65.4 | 59.4 | 52.0 | 49.2 |
| OPEN-SOURCE (2D) | NVILA-8B | 46.5 | 52.0 | 46.0 | 56.1 | 50.0 | 50.0 | 47.5 | 55.0 | 50.4 | 77.8 | 80.5 | 78.4 | 21.9 | 19.0 | 16.0 | 49.8 |
| | PALIGEMMA | 43.8 | 45.1 | 44.2 | 45.7 | 42.3 | 43.6 | 42.5 | 46.1 | 44.2 | 76.0 | 81.9 | 87.6 | 16.8 | 14.1 | 16.0 | 49.5 |
| | QWEN2.5-VL-3B | 51.3 | 54.2 | 57.1 | 58.8 | 54.5 | 59.6 | 59.6 | 66.7 | 57.7 | 24.0 | 17.3 | 12.1 | 82.9 | 84.4 | 83.1 | 49.9 |
| | QWEN2.5-VL-7B | 53.1 | 55.4 | 55.0 | 57.3 | 55.2 | 56.1 | 56.0 | 56.5 | 55.6 | 93.1 | 89.1 | 85.0 | 8.3 | 11.9 | 13.6 | 49.8 |
| | INTERNVL-8B | 47.7 | 50.6 | 49.4 | 47.9 | 47.1 | 48.7 | 48.9 | 51.0 | 48.9 | 98.8 | 99.5 | 99.7 | 1.5 | 2.2 | 2.2 | 50.6 |
| | DEEPSEEK-VL2 | 64.4 | 66.9 | 65.2 | 67.8 | 67.0 | 69.1 | 77.4 | 81.0 | 69.9 | 76.0 | 82.4 | 87.6 | 17.1 | 14.1 | 16.0 | 49.7 |
| | LLAVA-ONEVIS | 54.8 | 55.4 | 53.5 | 49.8 | 52.1 | 50.6 | 52.7 | 50.4 | 52.5 | 73.7 | 80.0 | 86.4 | 19.2 | 13.8 | 18.5 | 49.3 |
| OPEN-SOURCE (3D) | 3D-LLM | 27.0 | 28.0 | 30.5 | 30.3 | 33.9 | 32.5 | 36.8 | 37.7 | 32.1 | 86.8 | 90.5 | 84.3 | 7.5 | 9.6 | 7.6 | 47.7 |
| | LL3DA | 36.0 | 33.9 | 32.9 | 34.5 | 36.9 | 33.3 | 39.6 | 36.0 | 35.4 | 24.0 | 17.6 | 12.7 | 83.2 | 85.6 | 83.7 | 50.4 |
| | CHAT-SCENE | 30.8 | 32.5 | 40.7 | 49.0 | 41.6 | 45.3 | 43.3 | 51.0 | 41.8 | 24.3 | 18.1 | 12.4 | 82.9 | 85.3 | 83.4 | 50.3 |
| | 3D-LLAVA | 43.2 | 46.1 | 46.4 | 47.9 | 49.6 | 48.8 | 48.0 | 47.9 | 47.2 | 11.3 | 11.8 | 12.4 | 86.2 | 86.5 | 85.9 | 48.1 |
| | VIDEO-3D LLM | 50.0 | 50.2 | 50.7 | 51.1 | 49.4 | 50.4 | 51.1 | 49.3 | 50.3 | 25.2 | 22.5 | 20.8 | 74.6 | 79.9 | 78.3 | 50.2 |
| OURS | EGOTSR-COT | 68.3 | 70.2 | 72.2 | 73.7 | 74.5 | 75.6 | 76.7 | 77.8 | 73.6 | 91.9 | 90.1 | 93.0 | 1.8 | 2.2 | 2.6 | 46.9 |
| | EGOTSR-TAG | **82.4** | 84.8 | 86.2 | 87.1 | 87.7 | **88.6** | 88.6 | 88.7 | 86.8 | 55.0 | 49.0 | 58.7 | 62.3 | 54.9 | 54.9 | 55.8 |
| | **EGOTSR-LONGTAG** | 80.8 | **86.3** | **89.7** | **88.8** | **89.8** | 87.2 | 89.0 | 88.2 | **87.5** | 88.2 | 92.9 | 96.2 | **92.0** | **92.7** | **92.3** | **92.4** |

### 3.3.2. CHRONOLOGICAL BIAS

Traditional temporal training paradigms often cause models to fall into a pervasive "chronological bias" (Geirhos et al., 2020). Specifically, because training data is typically arranged in chronological order, models tend to learn superficial statistical regularities—implicitly assuming that the latter image in a sequence represents the completed task state. Consequently, during inference, the model may heuristically favor the second frame as being "closer to completion" without engaging in genuine logical reasoning. To mitigate this chronological bias in temporal reasoning (Liu et al., 2024), we implement a bidirectional evaluation methodology that includes both forward and inverse tasks.

Formally, let $Acc_{fwd}$ be the accuracy on the forward tuple $(I_a, I_b, \mathcal{L}_{task})$, and $Acc_{inv}$ be the accuracy on the inverse tuple $(I_b, I_a, \mathcal{L}_{task})$. A robust model must satisfy two conditions simultaneously:

$$
\begin{cases}
\text{High Average Accuracy:} & \frac{Acc_{fwd}+Acc_{inv}}{2} \to 1 \\
\text{Low Chronological Bias:} & |Acc_{fwd} - Acc_{inv}| \to 0
\end{cases}
\tag{12}
$$

This metric serves as the core "ruler" in our experiments to verify if the curriculum learning paradigm successfully decouples causal reasoning from temporal sequence shortcuts.

## 4. Experiment

### 4.1. Implementation Details

Our proposed EgoTSR model is fine-tuned based on the Qwen-VL-7B architecture. The training process strictly adheres to a three-stage curriculum learning paradigm, sequentially injecting CoT (spatial perception), Tag (state discrimination), and LongTag (logical planning) data. We conducted the distributed training on NVIDIA H800 GPUs and employed bfloat16 (bf16) mixed-precision training.

We use the AdamW optimizer with a peak learning rate of 2e-7 and a weight decay of 0.01. A cosine learning rate scheduler is applied, coupled with a 1,000-step linear warmup. The per-device batch size is set to 4, with a gradient accumulation of 2 steps.

## 4.2. Main Results

To rigorously validate the proposed approach, we conducted a systematic evaluation of current mainstream VLMs using our dual-level evaluation framework. As detailed in Table 1, this comprehensive benchmark compares our model against a diverse set of baselines—ranging from human perception and closed-source APIs to open-source 2D/3D models—across both short-horizon and long-horizon reasoning tasks. More model evaluation and deployment are shown in Appendix C.

On short-range tasks, the EgoTSR-CoT model demonstrated preliminary logical reasoning, with accuracy stabilizing around 70%, while EgoTSR-Tag, leveraging label-supervised data, boosted this intuitive perception to over 80%. However, despite its excellence in local perception, EgoTSR-Tag encountered significant bottlenecks in long-horizon tasks, where average accuracy fell below 60%, suggesting that mastering local spatial features alone is insufficient for multi-step logical closures.

Models like InternVL-8B show a drastic gap on Long tasks (>90% forward vs. <10% reverse), suggesting they exploit temporal priors and assume input order implies chronological precedence rather than performing scene analysis.

Addressing these limitations, the EgoTSR-36k model, strengthened by the third stage of long-tag data, achieved a qualitative breakthrough. Accuracy on long-horizon tasks surged from approximately 50% to over 90% (Avg: 92.4%), exhibiting exceptional robustness in both Forward and Inverse logical tests. Notably, this enhancement demonstrated an "incremental without degradation" characteristic; the model avoided catastrophic forgetting, maintaining high performance on short-range tasks (≈88%). These results validate that our progressive evolutionary strategy—advancing from Foundational Perception to Global Planning—effectively acquires high-level logical planning skills while consolidating foundational spatial perception.

## 4.3. The Trade-off between Spatial Perception and Logical Analysis

During training, as the number of training steps increases, the model steadily improves its long-sequence planning capabilities while maintaining the integrity and robustness of its atomic perception. As illustrated in Figure 3, the accuracy on long-horizon logical reasoning tasks significantly climbed from 74.3% at the early stages of training to 92.4%, demonstrating the model's increasingly sophisticated mastery of complex dependencies, progressively approaching human-level performance.

Concurrently, the model's performance on atomic short-term tasks was neither compromised nor inhibited by the optimization of high-level capabilities, remaining consistently

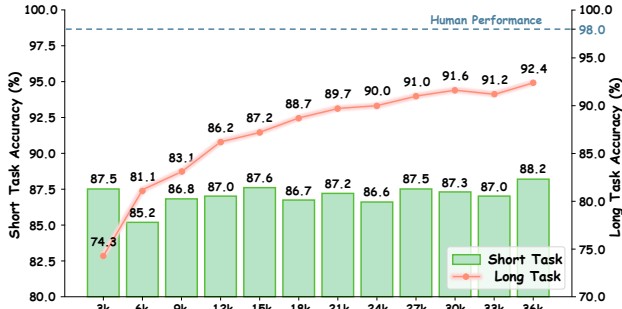

*Figure 3.* The dual-axis plot quantitatively demonstrates the efficacy of the training **Stage 3: LongTag**. The Long Task Accuracy (Red Line) exhibits a steep monotonic ascent, surging from an initial 74.3% to a peak of 92.4%. At the same time, the Short Task Accuracy (Green Bars) demonstrates remarkable stability, oscillating narrowly between 86.6% and 88.7%. This confirms that the model acquires complex planning capabilities while maintaining robust foundational spatial perception, effectively mitigating catastrophic forgetting.

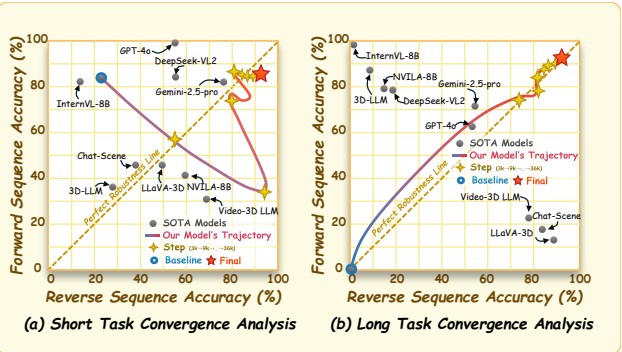

*Figure 4.* **Analysis of Oscillatory Convergence in Bidirectional Reasoning.** We visualize the training trajectories against the Perfect Robustness Line ($y = x$). (a) In **short tasks**, the model overcomes the initial "unidirectional reliance", fluctuating towards a balanced equilibrium. (b) In **long tasks**, the trajectory demonstrates learning resilience, evolving from a near-zero baseline to deep bidirectional alignment. Overall, the results verify that our paradigm effectively breaks the inherent **Chronological Bias**, pushing the model to the top-right high-performance quadrant.

stable at approximately 87.5% within a high-performance range. This trend indicates that our training paradigm effectively overcomes the risk of "catastrophic forgetting" commonly encountered in complex task learning. It achieves a significant leap in high-level logical planning without sacrificing foundational perceptual precision, ensuring a balanced evolution of embodied intelligence.

## 4.4. Analysis of Oscillatory Convergence in Bidirectional Reasoning

During the curriculum learning process, we observed a distinct trend of oscillatory convergence in model performance across both forward and inverse input tasks.

For short-sequence tasks (as illustrated in Figure 4 (a)), the

*Table 2.* **Quantitative ablation study of the EgoTSR framework.** The results illustrate the trade-off between Average Accuracy and the Bidirectional Gap. The transition from individual stages (S1, S2) to the full sequential curriculum (S1+S2+S3)—which significantly outperforms the non-sequential mixed training baseline (S1-S2-S3)—validates the effectiveness of our paradigm in mitigating chronological bias. Additionally, the Subtask Planner provides essential logical rigor to maximize accuracy.

| METHOD | FORWARD | INVERSE | AVG(↑) | GAP(↓) |
|---|---|---|---|---|
| EFFECTIVENESS OF CURRICULUM LEARNING | | | | |
| S1 | 91.7 | 2.2 | 46.9 | 89.5 |
| S1+S2 | 54.2 | 57.3 | 55.8 | 3.1 |
| S2 | 50.4 | 53.2 | 51.8 | 2.8 |
| S1-S2-S3 | 67.2 | 72.1 | 69.6 | 4.9 |
| S1+S2+S3 | **92.4** | **92.3** | **92.4** | **0.1** |
| EFFECTIVENESS OF SUBTASK PLANNER | | | | |
| W/O PLANNER | 83.2 | 78.9 | 81.1 | 4.3 |
| W/ PLANNER | **92.4** | **92.3** | **92.4** | **0.1** |

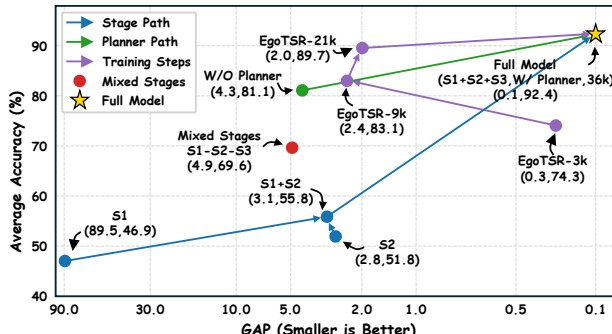

*Figure 5.* **Ablation Trajectories.** The plot visualizes the variation curve of Curriculum Learning Paradigm stages (Blue), Subtask Planner (Green), Mixed Stages (Red), and our EgoTSR models with different training steps (Purple) converge to the Full Model (Gold Star), maximizing Accuracy while minimizing the Gap and verifying the superiority of our structured evolutionary strategy.

base model (Qwen2.5-VL-7B) exhibits a severe "capability bias". While its forward accuracy exceeds 80%, its inverse accuracy remains below 30%. This significant imbalance reveals the model's unidirectional reliance on the sequential logic acquired during pre-training. However, as the gradient of fine-tuning data increases, the performances in both directions begin to equilibrate amidst fluctuations. Ultimately, the overall accuracy surges, converging toward the "perfect robustness line" (the $y = x$ diagonal).

For the more complex long-sequence tasks (as shown in Figure 4 (b)), the base model is virtually ineffective, residing near the origin of the coordinate system. Nevertheless, with the continuous injection of curriculum learning data, the model demonstrates remarkable learning resilience. Its performance trajectory steadily climbs from the bottom-left toward the top-right quadrant. Finally, the model not only achieves high-level accuracy under bidirectional inputs but also resides extremely close to the diagonal balance line.

These empirical results provide compelling evidence that incorporating high-quality, long-label data effectively breaks the **Chronological Bias** inherent in pre-training. This achieves deep bidirectional alignment, thereby significantly enhancing both the model's robustness and its precision in spatiotemporal reasoning.

### 4.5. Ablation Study

To validate EgoTSR, we analyzed the trade-off between Accuracy and the Chronological Bias Gap (Table 2 and Figure 5). The initial S1 stage suffers from severe bias (Gap 89.5), which, when integrating Tag data (S1+S2), reduces to 3.1, confirming the efficacy of our "explicit-to-internalized" strategy. Notably, our sequential curriculum (S1+S2+S3) significantly outperforms the mixed baseline (S1-S2-S3) (Gap: 0.1 vs. 4.9; Acc: 92.4% vs. 69.6%), underscoring

that ordered progression is indispensable for mitigating bias.

Furthermore, explicit task decomposition proves essential for long-horizon logic. While the base model achieves 81.1% accuracy without the planner, activating the Subtask Planner acts as a logical stabilizer, boosting accuracy to 92.4% and reducing the gap to 0.1.

Ultimately, the Curriculum Learning Paradigm eliminates bias (minimizing Gap), while the Subtask Planner is the decisive factor for mastering complex logic (maximizing Accuracy), jointly achieving robust consistency.

### 4.6. Case Study

To visually verify reasoning stability, we conducted comprehensive case studies across **Human Demonstrations**, **Simulations** (including *Behavior*, *Libero*, *Simpler*, and *RoboTwin*), and **Real-Robot Operations** (including *Franka Emika Panda*, *Agibot*, and *So-100* robotic arms). For each case, the model processes unsegmented long-horizon videos to generate real-time Task Completion curves.

As shown in Figure 6, for the long-horizon task "*Grab the green cup from the table and place it into the white plate*", the generated curve exhibits a distinct stepwise ascent aligning with atomic subtasks. Critical execution nodes (e.g., "Grab", "Place") trigger sharp gradients indicating high sensitivity to state transitions, while intermediate transport phases maintain stable slopes reflecting continuous spatial displacement. This robust alignment persists across diverse platforms, demonstrating that EgoTSR has successfully internalized a generalized "Perception-Planning-Decision" pathway. More case study results are shown in Appendix E.

## 5. Conclusion

We introduce EgoTSR, a framework addressing the issue of spatiotemporal reasoning. Leveraging EgoTSR-Data and

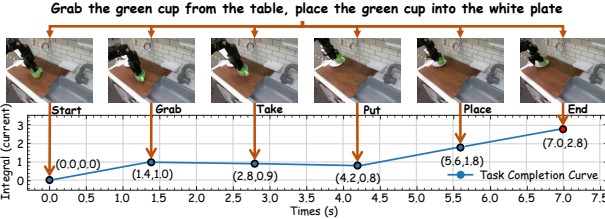

*Figure 6.* **Visualization of the Task Completion Progress Curve.** The figure aligns visual execution keyframes with the model's real-time inference. As the agent progresses through critical sub-goals, the Task Completion Curve exhibits a steady, monotonic ascent, accurately reflecting the accumulation of completed sub-tasks. This demonstrates our model's capability to perform fine-grained temporal monitoring across long-horizon sequences.

Curriculum Learning Paradigm, we guide the model from explicit CoT to intuitive Tag, and finally to LongTag. Furthermore, our Reasoning-Enhanced Task Decomposition bridges abstract goals and concrete execution. We establish a Dual-Level Evaluation Framework across short and long tasks. Extensive experiments show that EgoTSR outperforms state-of-the-art methods. By mitigating chronological bias, our work establishes a new standard for general-purpose robots in complex environments.

## Acknowledgements

We thank the anonymous reviewers for their constructive comments and valuable suggestions. We also thank our colleagues and collaborators for helpful discussions and feedback. This work was supported by National Natural Science Foundation of China under Grant No.U24A20326.

## Impact Statement

This paper presents work whose goal is to advance the field of Machine Learning. There are many potential societal consequences of our work, none which we feel must be specifically highlighted here.

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

# A. EgoTSR-Data Construction Pipeline

To support the proposed three-stage Curriculum Learning Paradigm, we construct the **EgoTSR-Data** (46M samples) starting from raw egocentric videos. The construction pipeline transforms continuous robotic manipulation trajectories into structured instruction-tuning samples, categorized into three subsets: $\mathcal{D}_{CoT}$, $\mathcal{D}_{Tag}$, and $\mathcal{D}_{Long}$. It ensures the model evolves from explicit reasoning to internalized intuition and finally to global planning. Sample data is shown in Figure 7.

## A.1. Data Source and Pre-processing

The foundation of our dataset is the **Agibot-World** dataset, which provides high-fidelity, large-scale egocentric video streams of real-world robotic operations.

**Multi-Scale Temporal Sampling:** To filter out high-frequency visual noise (e.g., camera jitter) and ensure significant state changes between frames, we first apply a uniform temporal down-sampling strategy ($\times 10$) to the raw video clips. Subsequently, to capture state changes across different granularities, we implement a multi-scale sampling strategy. Instead of relying solely on a fixed frame interval, we sample frame pairs $(I_a, I_b)$ with varying time gaps ($\Delta t$). Specifically, for short-term tasks, we sample intervals $\Delta t \in \{5, 6, \ldots, 11, 12+\}$ ($\times 10$), while for long-horizon tasks, sampling is divided into three levels - S, M, L, corresponding to INTRA-TASK, INTER-TASK, and MULTI-TASK in Equation 11. This approach allows the dataset to cover a wide spectrum of dynamics, ranging from short-range atomic movements to long-range task transitions.

**Bidirectional Frame Pair Extraction:** Based on the aforementioned sampling strategy, we extract paired frames from the processed sequences in a bidirectional manner. For each sampled pair, we construct both the forward tuple $(I_a, I_b)$ and the inverse tuple $(I_b, I_a)$. This bidirectional design is crucial for the "Dual-Level Evaluation Framework," forcing the model to discern temporal evolution solely from visual state changes rather than relying on the input order, effectively mitigating chronological bias.

## A.2. Multi-Granularity Annotation Generation

The core of the pipeline is the generation of annotations explicitly tailored to the three curriculum stages, utilizing a combination of automated rule-based extraction and the *Reasoning-Enhanced Task Decomposition* mechanism (described in subsection 3.2).

**Stage 1: Chain-of-Thought Data Construction** ($\mathcal{D}_{CoT}$). Focusing on *Knowledge Guidance* for the initial 15M samples, we generate detailed **Reasoning Paths** ($\mathcal{C}_{CoT}$) for each frame pair. By explicitly describing the spatial relationship between the robotic arm and the object before concluding the task state, we force the model to establish an explicit "Perception-Planning-Decision" pathway.

**Stage 2: Tag Data Construction** ($\mathcal{D}_{Tag}$). For the subsequent 16M samples, we transition to *Capability Internalization* by applying an **Information Pruning** process where intermediate textual reasoning chains are removed ($\mathcal{C} \to \emptyset$). Retaining only the visual input and ground truth label $y_{gt}$, this weak supervision signal compels the model to map visual observations directly to intuitive judgments, thereby reducing reliance on lengthy textual descriptions.

**Stage 3: LongTag Data Construction** ($\mathcal{D}_{Long}$). The final 15M samples target *Generalization Consolidation* for long-horizon planning. Instead of relying on Large Language Model's generation, we first extract ground-truth subtask sequences from the dataset metadata to formulate the Logical Skeleton $\mathcal{S}$ of orthogonal atomic subtasks. And then we train a Subtask Planner for more samples. To prevent overfitting and ensure diverse logical dependencies, we implement a **Triple-Level Sampling Mechanism** based on the subtask index mapping $\phi(I)$. This mechanism retrieves frame pairs spanning single steps (INTRA-TASK), adjacent transitions (INTER-TASK), and non-adjacent sequences (MULTI-TASK), with the ground-truth $\mathcal{S}$ injected into the prompt to provide strong supervision for verifying logical consistency across extended temporal spans.

## A.3. Standardization and Formatting

Finally, all processed samples were serialized into a unified JSON format. Each training sample is formulated as a tuple $x_i = (I_a, I_b, \mathcal{L}_{task}, y_{gt}, \mathcal{C})$, where the context $\mathcal{C}$ dynamically adapts (Reasoning Path, Empty, or Logical Skeleton) according to the specific curriculum stage.

# B. Standardized Prompts and Logical Templates

To mitigate the impact of prompt bias on evaluation outcomes, we designed structured templates for both short-term and long-horizon tasks.

### B.1. Short-term Spatial Awareness Prompt

Short-horizon evaluations are designed to induce a deep analysis of spatial state transitions through fine-grained textual feedback. The core prompt template is structured as follows:

Role: You are a spatial awareness expert.

Input Image: [Image 1], [Image 2],

Task Name: {task_name},

Instruction: Please observe the contents of the two images and carefully compare the differences between the two states.

Note: The two states have a chronological relationship (the exact order is not specified). The robotic arm in the images is performing the task: {task_name}. Please analyze and determine which of img1 or img2 is closer to completing the task. Provide a detailed and unambiguous analysis. Avoid using vague concepts (e.g., "fruit area") and focus strictly on the state of the robotic arm and the target objects involved in the task.

Output format: closer to completion: [img1 or img2].

### B.2. Long-horizon Planning Prompt

Long-horizon evaluations introduce sequences of atomic operations derived from task decomposition, requiring the model to achieve alignment across higher-dimensional temporal logic. The core prompt template is structured as follows:

Role: You are a spatial awareness expert.

Input Image: [Image 1], [Image 2],

Task Name: {task_name},

Subtasks: {subtask_sequence}.

Instruction: Your task is to observe and describe the two pictures (img1 and img2) respectively.

Hint: These two pictures occur in the same task.

Output format: closer to completion: [img1 or img2].

# C. Model Evaluation and Deployment Details

To ensure the impartiality and reproducibility of our evaluation results, we established a standardized deployment and testing pipeline tailored for Vision-Language Models (VLMs) with diverse architectures.

### C.1. Closed-Source Models (API-based)

For proprietary, closed-source models such as GPT-4o-mini, GPT-o4, and Gemini-2.5-Pro, we conducted evaluations using their official APIs. To minimize stochasticity in model responses and ensure deterministic outputs, all tests were performed with greedy decoding and a temperature of 0.

### C.2. Open-Source 2D Models (Image-Text Input)

For open-source 2D models, including PaliGemma, DeepSeek-VL2, InternVL-8B, and LLaVA-OneVision, we deployed them on a high-performance workstation equipped with dual NVIDIA RTX 3090 GPUs (totaling approximately 48GB of VRAM). This hardware configuration provides sufficient capacity for the full-precision inference of models with about 7B parameters. The evaluation logic remains strictly consistent with that of EgoTSR: each model receives two images along with the task name and generates a judgment based on a specific prompt.

### C.3. Open-Source 3D Models (Point Cloud-Text Input)

Furthermore, for models with native support for 3D point cloud inputs, such as LL3DA and Chat-Scene, we introduced a specialized pseudo-point-cloud restoration pipeline to bridge the modality gap, since our benchmark is based on egocentric 2D video streams. This process first employs a pre-trained depth estimation algorithm to obtain pixel-level depth maps, which are subsequently projected into 3D space using camera intrinsic parameters to generate semantic point clouds. Finally, the reconstructed 3D point clouds are combined with task instructions as input to the 3D-VLMs. This approach ensures that these models can effectively participate in the cross-model evaluation of egocentric spatio-temporal reasoning capabilities, even in the absence of native 3D sensor data.

## D. Final Evaluation Results

To verify the effectiveness of our proposed curriculum learning paradigm and reasoning-enhanced task decomposition mechanism, we conducted a systematic evaluation of current mainstream VLMs using a dual-level framework. The primary objective is to validate whether the models can balance the precision of micro-level spatial perception with the rigor of macro-level temporal planning when processing first-person perspective tasks.

The evaluation is categorized into two dimensions:

**Short-horizon Evaluation:** Utilizing a test set constructed from CoT data, this dimension emphasizes the model's sensitivity to spatial state changes within atomic actions. It aims to demonstrate that the model maintains a "high-fidelity" understanding of environmental feedback while enhancing its long-range reasoning capabilities.

**Long-horizon Evaluation:** Utilizing Long-Tag data, this dimension focuses on testing logical coherence when facing complex, multi-step tasks. It requires the model to accurately identify the completion status of each atomic step across extended temporal spans.

The following tables present the comprehensive testing results for human perception, closed-source 2D models, open-source 2D models, open-source 3D models, and our proposed model on both short-horizon and long-horizon tasks.

Complete test results are shown in Table 3.

## E. Case Study

In this section, we present comprehensive qualitative results of our case studies. To visually verify the reasoning stability and robustness of the proposed model, we conducted extensive evaluations across three primary domains:

- **Human Demonstrations:** Validating the model's ability to interpret diverse human motion patterns.

- **Simulated Environments:** Including *Behavior*, *Libero*, *Simpler*, and *RoboTwin*, providing a controlled setting for multi-task scenarios.

- **Real-World Robotic Platforms:** Leveraging *Franka Emika Panda*, *Agibot*, and *So-100* robotic arms to test deployment feasibility in physical environments.

For each case, the model processes unsegmented, long-horizon video streams to generate real-time **Task Completion Curves**, as shown in Figure 8. These curves serve as a continuous metric to verify the consistency of the model's temporal reasoning.

*Table 3.* The complete performance evaluation on **Dual-Level Evaluation Framework**. More EgoTSR models with fewer training steps are included. The left section details performance on the **Short** level. The right section details consistency on the **Long** level. **Bold** and underlined denote the best and second-best performance, respectively (excluding Huamn Perception).

| | MODEL | LEVEL 1: SHORT (ACC %) | | | | | | | | | LEVEL 2: LONG (ACC %) | | | | | | |
| --- | --- | --- | --- | --- | --- | --- | --- | --- | --- | --- | --- | --- | --- | --- | --- | --- | --- |
| | | FRAME INTERVAL (×10) | | | | | | | | AVG. | FORWARD | | | INVERSE | | | AVG. |
| | | 5 | 6 | 7 | 8 | 9 | 10 | 11 | ≥12 | | S | M | L | S | M | L | |
| | HUMAN PERCEPTION | 97.3 | 97.3 | 98.2 | 99.1 | 99.1 | 100.0 | 100.0 | 100.0 | 98.9 | 94.3 | 95.5 | 95.6 | 93.9 | 96.1 | 95.9 | 95.2 |
| CLOSE-SOURCE | GPT-O4MINI | 70.0 | 70.3 | 72.8 | 67.7 | 60.0 | 67.5 | 67.7 | 70.3 | 68.3 | 64.9 | 69.0 | 85.7 | 34.6 | 30.3 | 42.5 | 55.0 |
| | GPT-4O | 71.1 | 71.7 | 76.7 | 77.8 | 79.2 | 79.2 | 79.2 | 79.2 | 76.8 | 56.6 | 61.2 | 69.7 | 57.3 | 51.5 | 54.6 | 58.2 |
| | GPT-O3 | 67.2 | 61.4 | 59.6 | 61.6 | 59.6 | 64.0 | 62.0 | 66.6 | 62.8 | 71.9 | 64.7 | 73.1 | 69.6 | 66.7 | 68.6 | 69.0 |
| | GEMINI-2.5-PRO | 71.3 | 75.0 | 72.8 | 75.0 | 77.2 | 80.9 | 89.7 | **100.0** | 80.2 | 74.1 | 78.1 | 64.0 | 63.2 | 42.1 | 57.5 | 62.0 |
| | DOUBAO-1.5-PRO | 64.0 | 76.0 | 56.0 | 46.0 | 56.9 | 86.0 | 90.0 | 96.0 | 71.4 | 68.4 | 58.3 | 58.8 | 45.2 | 48.2 | 29.4 | 52.0 |
| | SEED-1.6 | 66.2 | 66.8 | 70.6 | 73.3 | 83.2 | 82.9 | 86.8 | 98.3 | 78.5 | 46.5 | 46.3 | 28.6 | 65.4 | 59.4 | 52.0 | 49.2 |
| OPEN-SOURCE (2D) | NVILA-8B | 46.5 | 52.0 | 46.0 | 56.1 | 50.0 | 50.0 | 47.5 | 55.0 | 50.4 | 77.8 | 80.5 | 78.4 | 21.9 | 19.0 | 16.0 | 49.8 |
| | PALIGEMMA | 43.8 | 45.1 | 44.2 | 45.7 | 42.3 | 43.6 | 42.5 | 46.1 | 44.2 | 76.0 | 81.9 | 87.6 | 16.8 | 14.1 | 16.0 | 49.5 |
| | QWEN2.5-VL-3B | 51.3 | 54.2 | 57.1 | 58.8 | 54.5 | 59.6 | 59.6 | 66.7 | 57.7 | 24.0 | 17.3 | 12.1 | 82.9 | 84.4 | 83.1 | 49.9 |
| | QWEN2.5-VL-7B | 53.1 | 55.4 | 55.0 | 57.3 | 55.2 | 56.1 | 56.0 | 56.5 | 55.6 | 93.1 | 89.1 | 85.0 | 8.3 | 11.9 | 13.6 | 49.8 |
| | INTERNVL-8B | 47.7 | 50.6 | 49.4 | 47.9 | 47.1 | 48.7 | 48.9 | 51.0 | 48.9 | **98.8** | **99.5** | **99.7** | 1.5 | 2.2 | 2.2 | 50.6 |
| | DEEPSEEK-VL2 | 64.4 | 66.9 | 65.2 | 67.8 | 67.0 | 69.1 | 77.4 | 81.0 | 69.9 | 76.0 | 82.4 | 87.6 | 17.1 | 14.1 | 16.0 | 49.7 |
| | LLAVA-ONEVIS | 54.8 | 55.4 | 53.5 | 49.8 | 52.1 | 50.6 | 52.7 | 50.4 | 52.5 | 73.7 | 80.0 | 86.4 | 19.2 | 13.8 | 18.5 | 49.3 |
| OPEN-SOURCE (3D) | 3D-LLM | 27.0 | 28.0 | 30.5 | 30.3 | 33.9 | 32.5 | 36.8 | 37.7 | 32.1 | 86.8 | 90.5 | 84.3 | 7.5 | 9.6 | 7.6 | 47.7 |
| | LL3DA | 36.0 | 33.9 | 32.9 | 34.5 | 36.9 | 33.3 | 39.6 | 36.0 | 35.4 | 24.0 | 17.6 | 12.7 | 83.2 | 85.6 | 83.7 | 50.4 |
| | CHAT-SCENE | 30.8 | 32.5 | 40.7 | 49.0 | 41.6 | 45.3 | 43.3 | 51.0 | 41.8 | 24.3 | 18.1 | 12.4 | 82.9 | 85.3 | 83.4 | 50.3 |
| | 3D-LLAVA | 43.2 | 46.1 | 46.4 | 47.9 | 49.6 | 48.8 | 48.0 | 47.9 | 47.2 | 11.3 | 11.8 | 12.4 | 86.2 | 86.5 | 85.9 | 48.1 |
| | VIDEO-3D LLM | 50.0 | 50.2 | 50.7 | 51.1 | 49.4 | 50.4 | 51.1 | 49.3 | 50.3 | 25.2 | 22.5 | 20.8 | 74.6 | 79.9 | 78.3 | 50.2 |
| OURS | EGOTSR-COT | 68.3 | 70.2 | 72.2 | 73.7 | 74.5 | 75.6 | 76.7 | 77.8 | 73.6 | 91.9 | 90.1 | 93.0 | 1.8 | 2.2 | 2.6 | 46.9 |
| | EGOTSR-TAG | **82.4** | 84.8 | 86.2 | 87.1 | 87.7 | 88.6 | 88.6 | 88.7 | 86.8 | 55.0 | 49.0 | 58.7 | 62.3 | 54.9 | 54.9 | 55.8 |
| | EGOTSR-3K | 82.2 | 83.9 | 88.2 | 87.3 | 87.0 | 84.7 | 86.6 | 87.5 | 85.9 | 70.0 | 75.9 | 76.5 | 76.6 | 72.5 | 74.2 | 74.3 |
| | EGOTSR-9K | 80.7 | 85.3 | 85.4 | 87.8 | 89.2 | 82.6 | 82.4 | 86.8 | 85.0 | 75.7 | 82.4 | 87.6 | 83.2 | 85.8 | 84.0 | 83.1 |
| | EGOTSR-15K | 76.9 | 84.4 | 88.2 | 85.2 | 88.1 | 86.7 | 85.9 | 87.6 | 86.1 | 82.1 | 89.0 | 91.1 | 85.6 | 88.0 | 87.2 | 87.2 |
| | EGOTSR-21K | 79.3 | 85.3 | 87.3 | 87.8 | 87.5 | **89.3** | 87.2 | 87.2 | 87.3 | 81.6 | 90.1 | 94.3 | 89.5 | 91.4 | 91.0 | 89.7 |
| | EGOTSR-27K | 79.8 | **87.8** | 87.3 | 87.8 | 88.1 | 86.7 | 87.2 | 87.5 | **87.5** | 81.6 | 90.0 | 92.1 | **92.5** | **95.7** | **93.8** | 91.0 |
| | EGOTSR-33K | 77.4 | 84.9 | 87.3 | **88.8** | 89.8 | 86.2 | 88.4 | 87.0 | 87.4 | 84.7 | 92.1 | 93.3 | 90.4 | 94.5 | 92.0 | 91.2 |
| | **EGOTSR-LONGTAG** | 80.8 | 86.3 | **89.7** | **88.8** | 89.8 | 87.2 | 89.0 | 88.2 | **87.5** | 88.2 | 92.9 | **96.2** | 92.0 | 92.7 | 92.3 | **92.4** |

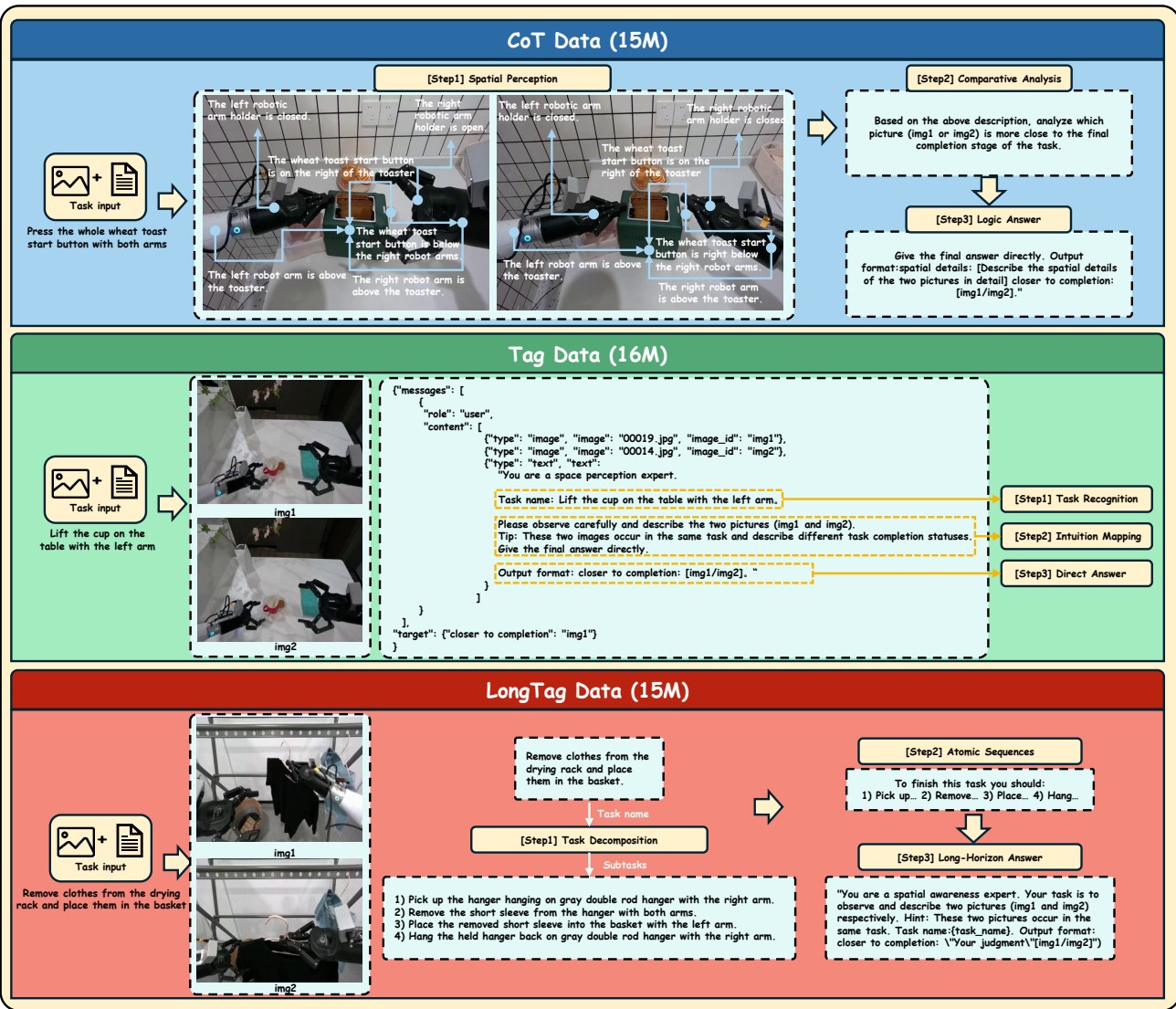

*Figure 7.* **Sample data.** This figure shows the detailed structure and examples of three types of data corresponding to our Curriculum Learning Paradigm: CoT Data (Blue) to establish explicit reasoning chains, Tag Data (Green) to foster internalized perception, and LongTag Data (Red) to enable complex decision-making for long-horizon tasks.

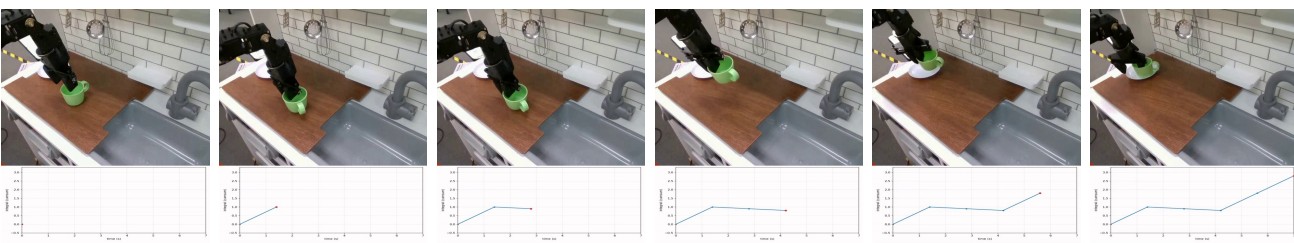

(a) Grab the green cup from the table, place the green cup into the white plate

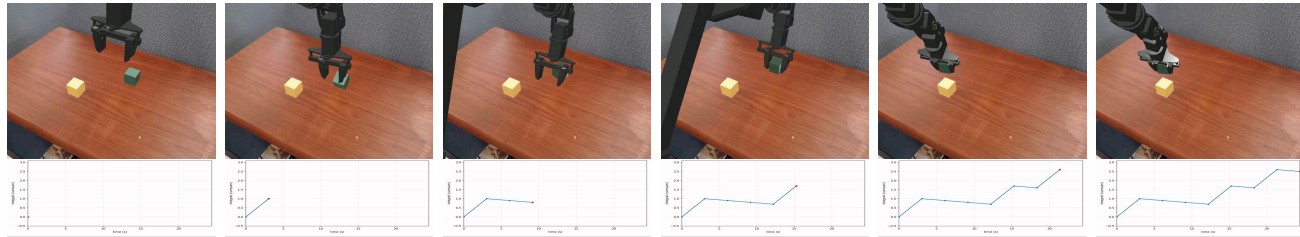

(b) Grab the green cube from the table, lift the green cube and place it on top of the yellow cube

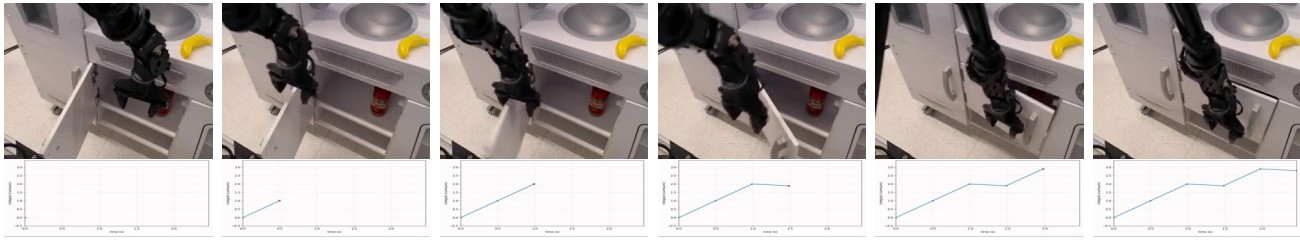

(c) Grab the white cabinet door, close the white cabinet door

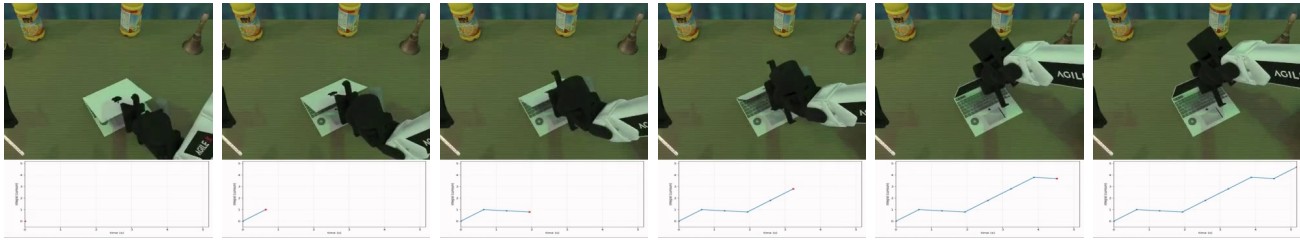

(d) Lift the laptop screen upward

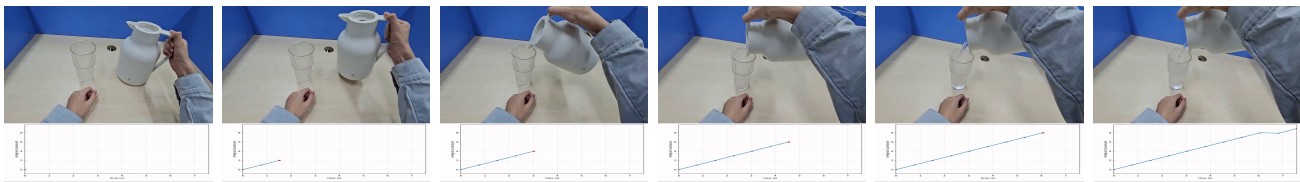

(e) Lift the kettle and pour water from the kettle into the cup

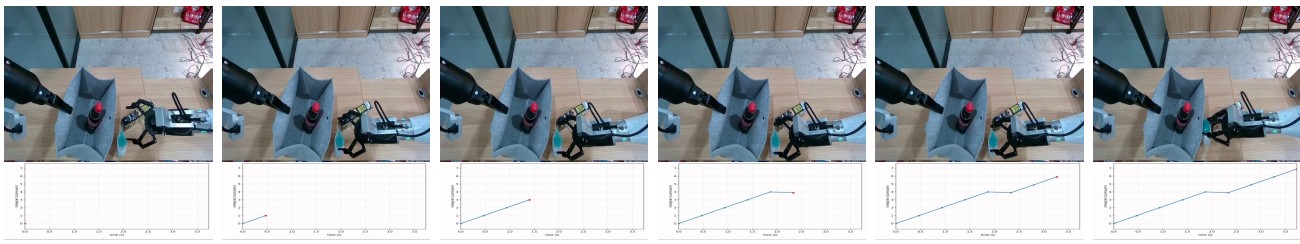

(f) Grab the hand sanitizer on the table with the right arm

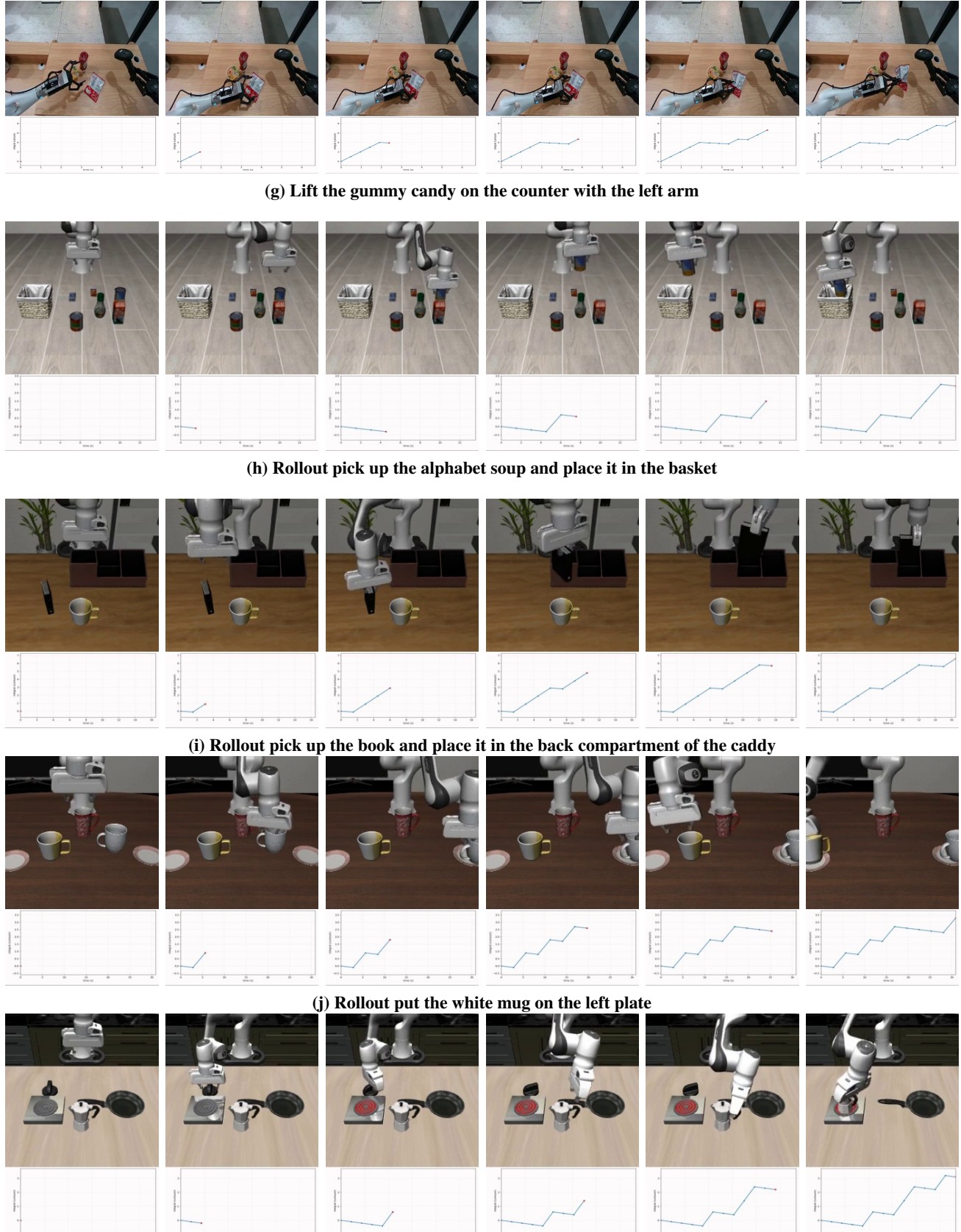

(g) Lift the gummy candy on the counter with the left arm

(h) Rollout pick up the alphabet soup and place it in the basket

(i) Rollout pick up the book and place it in the back compartment of the caddy

(j) Rollout put the white mug on the left plate

(k) Rollout turn on the stove

*Figure 8.* Robot manipulation case studies.

