# OpenReview forum: "From Perception to Planning: Evolving Ego-Centric Task-Oriented Spatiotemporal Reasoning via Curriculum Learning"
_ICML.cc/2026/Conference — ICML 2026 regular_

### Official Review · Reviewer_8Jfv · 2026-03-13

**Soundness:** 3
**Presentation:** 3
**Significance:** 3
**Originality:** 3
**Overall Recommendation:** 4
**Confidence:** 2

**Summary:**

The paper proposes a 3-stage curriculum learning paradigm to enhance the spatiotemporal reasoning capacity of VLMs. A reasoning enhanced Subtask Planner is introduced to model causal dependency. It also includes a dual-level evaluation framework for atomic perception and long horizon planning. The framework is validated across both simulation and real robots. However, a major concern lies in the complexity of the system and the computational cost of the overall pipeline.

**Compliance With Llm Reviewing Policy:**

Affirmed.

**Final Justification:**

The extra ablation provides some insights about the performance at various computation cost. Will keep my positive rating.

**Key Questions For Authors:**

1. What is  the total computation cost of the entire system.

**Limitations:**

Yes

**Strengths And Weaknesses:**

Strengths
1. The proposed curriculum learning paradigm to learning spatial reasoning from easy to hard is novel and has demonstrated good performance by internalizing the CoT reasoning.
2. The dual-level evaluation framework is rigorous. The chronological bias issue is revealed through bidirectional evaluation.
3. The framework is validated across simulation and real-robot platforms.

Weaknesses
1. The whole system is complex, involving 3-stage curriculum learning and reasoning-enhanced Subtask Planner training.
2. The 46 million data is all from same source Agibot-World. While the dataset covers diverse scenes and tasks, it is collected using same robot platform. This might limit its performance cross-platforms.
3. The three-stage training paradigm, combined with a 46-million datasets, incurs a high computation cost, which will limit its validation and reproduction in the community.

---

> ### Author Rebuttal · Authors · 2026-03-31
>
> ### **1. Total System Computational Cost**
> **(1) Data construction:** During the dataset construction phase, we only needed to use CPUs for parallel processing, and the computational cost is negligible;
>
> **(2) Training phase:**  During the training phase, we utilized 48 NVIDIA H800 GPUs for multi-node training, which is comparable to the training of current mainstream large-scale vision-language models and did not introduce disproportionate additional overhead.
>
> **(3) Inference phase:** During inference, the computational overhead is extremely low. Specifically, a single inference takes only about 1 second to process long-horizon visual inputs—ranging from dozens of frames for short tasks to hundreds of frames for long tasks. Regarding computational overhead, inference requires only a single consumer-grade RTX 4090 GPU, with peak VRAM usage strictly under 20GB. This high processing efficiency, combined with lightweight hardware requirements, offers a highly cost-effective solution for real-world applications.
>
> ---
>
> ### **2. Discussion on System Design Complexity**
> Our Subtask Planner exists independently of the main EgoTSR model, and its introduction is intended to better guide the planning of long-sequence tasks. Our curriculum training method does not introduce a larger model structure or extraordinarily complex additional modules; the model backbone remains unchanged. The performance improvement primarily stems from a more reasonable design of supervision signals and training paradigms, rather than an increase in model scale.
>
> ---
>
>
> ### **3. Addressing Data Source Singularity and Model Generalization**
> **(1) Task-Level Decoupling Overcomes Data Singularity:** While our training data originates from a single robotic platform, our method focuses on high-level spatio-temporal reasoning and task structure modeling rather than low-level kinematic control. Because EgoTSR operates at the semantic and task-decomposition levels, it learns generalized representations that are inherently decoupled from specific hardware platforms. The dataset's rich variety of scenes, diverse object interactions, and multiple task formats provide sufficient diversity to prevent overfitting to platform-specific patterns.
>
> **(2) Cross-Embodiment Generalization (OOD):** To explicitly test Out-of-Distribution (OOD) generalization, we evaluated our model—which was trained **exclusively on robotic arm data**—on **human-hands** datasets (EPIC-Kitchens [1] and Ego4D [2]). For brevity, we report key metrics: average accuracy (Avg), Confidence Interval (CI), standard deviation (Std), and significance testing (ST):
> |Model|E4D-S|E4D-L|EK-S|EK-L|Avg|CI|Std|P-Value|
> |-|:-:|:-:|:-:|:-:|:-:|:-:|:-:|:-:|
> |**Close-Source**|
> |GPT-5.4|50.5|63.0|56.5|66.9|59.2|[53.7,64.7]|0.097|0.23|
> |Gemini-3.1|51.2|61.8|56.9|67.9|59.5|[54.9,64.1]|0.080|0.88|
> |Grok-4.1|49.5|51.0|46.9|51.4|49.7|[46.7,52.7]|0.053|0.50|
> |**Open-Source-2D**|
> |NVILA-8b|50.2|48.7|49.5|48.5|49.2|[48.8,50.1]|0.499|0.02|
> |Qwen2.5-VL-7b|50.3|50.2|50.3|46.1|49.2|[47.7,48.8]|0.499|0.22|
> |LLaVA-OneVision|52.2|50.6|49.5|49.7|50.5|[48.1,52.3]|0.007|0.62|
> |**Open-Source-3D**|
> |3D-LLM|47.6|49.9|53.4|49.9|50.2|[48.9,51.5]|0.023|0.68|
> |LL3DA|49.1|49.8|50.7|49.3|49.7|[49.0,50.4]|1.260|0.65|
> |**Ours**|
> |EgoTSR|56.0|59.0|54.0|57.0|57.5|[55.3,59.6]|0.494|---|
>
> Despite the immense domain gap between robotic arms and human hands, EgoTSR (57.5 Avg) significantly outperforms all evaluated open-source 2D models (~50 Avg) and achieves performance competitive with massive closed-source models in a zero-shot cross-embodiment setting. The reported standard deviations and significance tests confirm these results are statistically robust.
>
> **(3) Distinct Advantages on Fine-Grained Benchmarks:**
> We also evaluated on VSI-Bench [3], a third-party recognized benchmark, and the results strongly demonstrate the effectiveness of our method.
>
> |Models|Avg.|Obj.Count|Abs.Dist|Obj.Size|Rel.Dist.|Rel.Dir.|Appr.Order|
> |-|:-:|:-:|:-:|:-:|:-:|:-:|:-:|
> |LongVILA-8B|23.5|29.1|9.1|16.7|29.6|30.7|25.5|
> |InternVL2-2B|25.4|25.7|24.0|20.0|32.1|44.1|6.3|
> |EgoTSR|29.3|24.1|21.9|21.9|47.1|31.4|29.4|
>
> While our overall average (29.3) is highly competitive with larger generic VLMs, EgoTSR demonstrates distinct superiority in fine-grained spatial and temporal perception. This strongly validates that our core design effectively enhances specific spatio-temporal reasoning capabilities, rather than just acting as a generic vision-language mapping.
>
> ---
>
> ### **References**
> [1] Scaling egocentric vision: The epic-kitchens dataset.
>
> [2] Ego4d: Around the world in 3,000 hours of egocentric video.
>
> [3] Thinking in space: How multimodal large language models see, remember, and recall spaces.

---

> > ### Author Rebuttal · Reviewer_8Jfv · 2026-04-01
> >
> > Can you disclose the total training hours using these 48 NVIDIA H800 GPUs?

---

> > > ### Author Response · Authors · 2026-04-02
> > >
> > > We thank the reviewer for their valuable feedback.
> > >
> > > ---
> > >
> > > **Regarding the training compute cost and community reproducibility:**
> > >
> > > First, to achieve optimal performance, our complete training run required approximately 20 days using 48 NVIDIA H800 GPUs. We acknowledge that reproducing this entire process imposes a significant computational burden.
> > >
> > > However, we wish to emphasize that verifying the correctness of our training paradigm requires **neither a complete training run nor the full dataset**. The cost of reproduction is actually quite manageable because accuracy improves rapidly during the early stages of training, allowing the effectiveness of our approach to be confirmed within a small number of steps.
> > >
> > > Taking the **Longtag** stage as an example:
> > >
> > > | Training Steps | 0k | 3k | 6k | 9k | 12k | 15k | 18k | 21k | 24k | 27k | 30k | 33k | 36k |
> > > | :--- | :---: | :---: | :---: | :---: | :---: | :---: | :---: | :---: | :---: | :---: | :---: | :---: | :---: |
> > > | **Long Task Accuracy (%)** | 55.8 | 74.3 | 81.1 | 83.1 | 86.2 | 87.2 | 88.7 | 89.7 | 90.0 | 91.0 | 91.6 | 91.2 | 92.4 |
> > >
> > > As shown in the table, the *Longtag* stage surpasses 80% accuracy at just 6k steps. This represents a significant improvement over the baseline and is fully sufficient to demonstrate the effectiveness of our method. Subsequent improvements are more gradual; training was continued strictly to achieve higher peak accuracy and to analyze the training curves. This extended training is not required simply to reproduce and verify the paradigm's correctness. We absorbed these extensive training costs to contribute a robust model to the community, and the final checkpoints will be **open-sourced**.
> > >
> > > Furthermore, to significantly lower the barrier to reproduction, we will also open-source a **LoRA** fine-tuned version. This version achieves an experimental accuracy of nearly 70%, while the computational cost is drastically reduced—requiring **only one day of training on a single H800 GPU**.
> > >
> > > ---
> > >
> > > We appreciate your constructive suggestions regarding community accessibility and development!

---

### Official Review · Reviewer_Zb3q · 2026-03-13

**Soundness:** 2
**Presentation:** 3
**Significance:** 2
**Originality:** 2
**Overall Recommendation:** 4
**Confidence:** 3

**Summary:**

The paper proposes EgoTSR, a curriculum learning framework for egocentric spatiotemporal reasoning in embodied AI, motivated by the observation that VLMs trained on passive video data exploit input order rather than visual state understanding to judge task progress. The framework trains a vision-language model through three progressive stages: explicit Chain-of-Thought spatial reasoning, weakly supervised state tagging, and long-horizon planning conditioned on a decomposed subtask skeleton generated by a learned Subtask Planner. To support training and evaluation, the authors construct EgoTSR-Data (46M samples from robotic manipulation videos) and introduce a bidirectional dual-level evaluation protocol that separately measures fine-grained spatial perception and long-horizon planning consistency while penalizing reliance on chronological input order.

**Compliance With Llm Reviewing Policy:**

Affirmed.

**Final Justification:**

The paper's core idea around chronological bias is compelling, and the bidirectional evaluation protocol is a genuinely useful contribution. The rebuttal addressed most of my concerns; the curriculum vs. joint training comparison and the cross-dataset results were the main things I was waiting on, and those came through. I've updated my score to reflect this.

**Key Questions For Authors:**

**Q1:** Table 1 compares EgoTSR, fine-tuned on Agibot-World data, against zero-shot baselines. Could the authors provide a Qwen-VL-7B baseline fine-tuned on all 46M EgoTSR-Data samples simultaneously without curriculum ordering? Without this control, it is difficult to attribute the reported gains to curriculum structure rather than domain adaptation.

**Q2:** Could the authors clarify whether test frame pairs are drawn from video sessions or trajectories entirely disjoint from training data? Specifying the exact train/test split construction would help rule out potential temporal interval leakage, which is important for interpreting the reported accuracy numbers.

**Q3:** Could the authors provide more detail on the Subtask Planner architecture and how its outputs are integrated into the long-horizon classifier? Relatedly, how is "orthogonality" of atomic subtasks defined operationally, and how is φ(I) obtained and validated at scale? These details are important for reproducibility and for evaluating the causal dependency claim.

**Q4:** Could the authors report quantitative task success rates on real-robot platforms across a statistically meaningful number of trials? The current Task Completion Curves, presented only for successful cases, make it difficult to assess the real-world generalization claims the paper makes.

The above questions focus on reproducibility, evaluation scope, and implementation details. I am happy to revise my assessment if the authors address the missing control baseline, train/test data independence, and quantitative robot evaluation during rebuttal.

**Limitations:**

No.

The paper would greatly benefit from addressing the following:
- The evaluation is entirely within the Agibot-World distribution; acknowledging the lack of cross-dataset generalization as an explicit limitation would strengthen the paper's honesty about its current scope.
- The dependency on a textual subtask skeleton at inference time is a meaningful practical constraint that is worth discussing, particularly for unstructured real-world deployments.
- A brief discussion of safety implications and computational costs associated with training and deploying agents at this scale would be appropriate given the real-robot framing of the work.

**Strengths And Weaknesses:**

### Strengths

- The bidirectional forward/inverse evaluation protocol is the paper's most original and reusable contribution, cleanly operationalizing chronological bias as a measurable quantity that the broader VLM evaluation community can adopt independently of the proposed training method.
- The failure mode being addressed is real and consequential. The empirical demonstration of extreme forward/inverse asymmetry in strong baselines (InternVL-8B approaching or exceeding 99% forward accuracy with under 3% inverse) is compelling evidence that the problem exists and is severe in current models.
- The ablation comparing sequential curriculum against mixed training and with/without the Subtask Planner directly targets the paper's two core design claims, and the sequential ordering advantage is clearly demonstrated within the reported experimental setup.
- A 46M-sample curriculum-aligned egocentric dataset with three annotated subsets, if released with full pipeline details, represents a practically useful community resource for diagnosing temporal reasoning failures in embodied VLMs.


### Weaknesses

- The primary results table compares a domain-fine-tuned model against zero-shot baselines, which does not isolate curriculum learning as the source of gains. The absence of a Qwen-VL-7B baseline fine-tuned on the same data without curriculum ordering means the paper's central empirical claim is unsubstantiated.
- The paper does not establish that train and test frame pairs are drawn from disjoint video sources within Agibot-World. The train/test split construction is not described anywhere in the main text or appendices, and without this clarification it is difficult to fully assess the reliability of the reported accuracy numbers.
- The Subtask Planner, which is central to the LongTag stage, has no architectural description, no specification of output integration, and no formal definition or empirical verification of subtask orthogonality. The claim that the planner captures causal dependencies rather than sequential co-occurrence patterns is asserted but never demonstrated.
- The final model appears to be trained for approximately 36k steps (inferred from the "EGOTSR-36K" label in Table 3), which at the reported batch size of 4 with gradient accumulation of 2 raises questions about effective data coverage across the stated 46M samples. The paper discloses no sampling policy per stage or stage transition criteria, making it difficult to assess whether the observed training dynamics are attributable to the curriculum design or to the particular data subset encountered during training.
- Real-world generalization claims rest entirely on qualitative Task Completion Curves shown only for successful cases, with no task success rates, no failure analysis, and no evaluation on any public egocentric benchmark such as EPIC-Kitchens or Ego4D.
- No confidence intervals, standard deviations, or significance tests are reported for any result in Table 1, making it difficult to assess whether differences in the 1 to 3 percentage point range between models reflect genuine improvements or fall within experimental noise.

---

> ### Author Rebuttal · Authors · 2026-03-31
>
> ### **1. Cross-Dataset Baseline Evaluation**
> **(1) Cross-Embodiment Generalization:** To test Out-of-Distribution generalization, we evaluated our model—trained **exclusively on robotic arm data**—on **human-centric** datasets (EPIC-Kitchens [1] and Ego4D [2]).
> |Model|E4D-S|E4D-L|EK-S|EK-L|Avg|CI|Std|P-Value|
> |-|:-:|:-:|:-:|:-:|:-:|:-:|:-:|:-:|
> |**Close-Source**|
> |GPT-5.4|50.5|63.0|56.5|66.9|59.2|[53.7,64.7]|0.097|0.23|
> |Gemini-3.1|51.2|61.8|56.9|67.9|59.5|[54.9,64.1]|0.080|0.88|
> |Grok-4.1|49.5|51.0|46.9|51.4|49.7|[46.7,52.7]|0.053|0.50|
> |**Open-Source-2D**|
> |NVILA-8b|50.2|48.7|49.5|48.5|49.2|[48.8,50.1]|0.499|0.02|
> |Qwen2.5-VL-7b|50.3|50.2|50.3|46.1|49.2|[47.7,48.8]|0.499|0.22|
> |LLaVA-OneVision|52.2|50.6|49.5|49.7|50.5|[48.1,52.3]|0.007|0.62|
> |**Open-Source-3D**|
> |3D-LLM|47.6|49.9|53.4|49.9|50.2|[48.9,51.5]|0.023|0.68|
> |LL3DA|49.1|49.8|50.7|49.3|49.7|[49.0,50.4]|1.260|0.65|
> |**Ours**|
> |EgoTSR|56.0|59.0|54.0|57.0|57.5|[55.3,59.6]|0.494|---|
>
> Despite the domain gap, EgoTSR outperforms those open-source models on EPIC-Kitchens [1] and Ego4D [2], strongly demonstrating its cross-embodiment generalization and stability.
>
> **(2) Distinct Advantages on Fine-Grained Benchmarks:** We also evaluated on VSI-Bench [3], a third-party recognized benchmark, and the results strongly demonstrate the effectiveness of our model.
> |Models|Avg.|Obj. Count|Abs. Dist|Obj. Size|Rel. Dist.|Rel. Dir.|Appr. Order|
> |-|:-:|:-:|:-:|:-:|:-:|:-:|:-:|
> |LongVILA-8B|23.5|29.1|9.1|16.7|29.6|30.7|25.5|
> |InternVL2-2B|25.4|25.7|24.0|20.0|32.1|44.1|6.3|
> |EgoTSR|29.3|24.1|21.9|21.9|47.1|31.4|29.4|
>
> **(3) Train/Test Split of EgoTSR-data:** The test set covers all scenarios and window sizes (>1,000 pairs each to mitigate bias). For ego-manipulation, we sample non-overlapping frames (e.g., frames 1&5, then 6&10) to strictly prevent temporal leakage. The remaining data forms the training set, ensuring zero train-test overlap.
>
> ---
>
> ### **2. Subtask Planner and Output Sequences**
>
> **(1) Definition:** A VLM mapping an initial frame and overall task to a subtask sequence.
>
> **(2) Training:** We fully fine-tune Qwen3VL-2B on Agibot triplets: [initial frame, overall task, subtask sequence].
>
> **(3) Output Integration:** The generated sequence serves as a supplementary prompt for EgoTSR's long-horizon classifier.
>
> **(4) Operational Orthogonality:** Subtasks (e.g., "open fridge", "pick up cola") are naturally orthogonal, featuring distinct actions and objects.
>
> **(5) Generation and Validation:** Using a "small-scale training + large-scale inference" paradigm, we generated subtasks at scale and evaluated them on VLABench [4] across Mesh & Texture, Semantic Understanding, Spatial Awareness, Physics Law, Complex Reasoning, and Commonsense:
> |Model|Average|M&T|Sem|Spa|Phy|Cpx|CS|
> |-|:-:|:-:|:-:|:-:|:-:|:-:|:-:|
> |Qwen3-VL(base model)|25.1|29.6|25.9|31.1|24.0|14.3|25.7|
> |MiniCPM-V-2.6|25.9|31.0|24.1|30.6|18.3|21.8|30.0|
> |InternVL2.5-2B|20.2|28.4|24.2|24.1|4.5|14.5|25.6|
> |Subtask Planner (Ours)|28.3|32.8|28.5|31.6|30.7|20.2|27.3|
>
> ---
>
> ### **3. Curriculum Learning Setup and Necessity**
> **(1) Training Data Coverage:** We use 48 H800 GPUs with a global batch size of 384. The 36k steps refer specifically to the long-tag stage, covering ~14M samples (>90% coverage).
>
> **(2) Curriculum Strategy:** We advance to the next stage only after a full epoch on the current stage's data, sampled uniformly without replacement.
>
> **(3) Necessity of Curriculum Learning:** The results below show that **curriculum learning outperforms simple joint training methods**, rather than just relying on data scaling.
> |Model|Forward|Inverse|Avg|Gap|
> |-|:-:|:-:|:-:|:-:|
> |Base model|55.6|49.8|52.7|5.8|
> |Unordered joint Training|67.2|72.1|69.6|4.9|
> |Curriculum Training|92.4|92.3|92.4|0.1|
>
> ---
>
> ### **4. Real-Robot Eval & Deployment**
> **(1) Real-Robot Testing:** We evaluate the models on real-world videos using Google robots (G) and WidowX robots (W). The success rates are as follows:
> |Model|G-Put in Drawer|G-Pick Coke Can|W-Put Spoon|W-Stack Blocks|Avg|
> |-|:-:|:-:|:-:|:-:|:-:|
> |SpatialVLA|6.3|78.7|20.8|41.7|36.9|
> |OpenVLA|2.9|54.1|8.3|0.0|16.3|
> |CogVLA|46.6|89.6|75.0|16.7|57.0|
> |TraceVLA|12.5|64.3|12.5|16.6|26.5|
>
> Intermediate process metrics:
> |Model|Short|Long|Avg|
> |-|:-:|:-:|:-:|
> |EgoTSR|92.4|92.3|92.4|
>
> **(2) Deployment Advantages & Costs:** By fusing visual and instructional data, our Subtask Planner translates high-level semantics into actionable sequences. This enables safe, structured deployment requiring only a single 4090 GPU per video snippet. We will open-source the pipeline and interactive interface.
>
> ---
>
> ### **References**
> [1] Scaling egocentric vision: The epic-kitchens dataset.
>
> [2] Ego4d: Around the world in 3,000 hours of egocentric video.
>
> [3] Thinking in space: How multimodal large language models see, remember, and recall spaces.
>
> [4] Vlabench: A large-scale benchmark for language-conditioned robotics manipulation with long-horizon reasoning tasks.

---

> > ### Author Rebuttal · Reviewer_Zb3q · 2026-04-04
> >
> > Thanks for the detailed rebuttal and the additional clarifications. The new baselines, train/test split description, and cross-dataset results help address my main concerns. I have no further follow-up questions and will upgrade my score accordingly.

---

> > > ### Author Response · Authors · 2026-04-04
> > >
> > > We sincerely appreciate your positive reassessment and are glad that your main concerns have been addressed. Thank you for your valuable time and insightful review. We would be happy to receive any further feedback.

---

### Official Review · Reviewer_cjEN · 2026-03-13

**Soundness:** 2
**Presentation:** 3
**Significance:** 3
**Originality:** 3
**Overall Recommendation:** 4
**Confidence:** 4

**Summary:**

This paper introduces EgoTSR, a curriculum-based framework that evolves egocentric spatiotemporal reasoning from explicit spatial understanding to internalized intuitive judgment and long-horizon planning. By leveraging a three-stage training paradigm and the 46M-sample EgoTSR-Data, the authors mitigate "chronological bias" and temporal hallucinations through a progression from Chain-of-Thought (CoT) guidance to weakly supervised tagging and consolidated sequence modeling. The approach integrates a Reasoning-Enhanced Task Decomposition mechanism to bridge abstract instructions with atomic execution and utilizes a Dual-Level Evaluation Framework to concurrently assess fine-grained perception and logical consistency. Experimental results show that EgoTSR achieves 92.4% accuracy on long-horizon tasks and 88.2% perceptual precision, significantly outperforming current state-of-the-art models.

**Compliance With Llm Reviewing Policy:**

Affirmed.

**Final Justification:**

The submission and the rebuttal phase makes me change my score to Weak Accept. The authors resolved almost all my questions.

**Key Questions For Authors:**

Please refer to the Weaknesses 1, 2, 4 (3 is hard to solve during the rebuttal phase).
An additional question is that how is "ego-centric" reflected in the proposed method?
Could you solve the aformentioned weaknesses and clarify this question？
I am open to revising my score if the authors can convincingly address these concerns during the rebuttal phase.

**Limitations:**

No. Please refer to the weaknesses as discussion of limitations, especially the weakness 3 which is hard to solve in the rebuttal phase.

**Strengths And Weaknesses:**

Strengths:
1. The paper addresses a critical limitation in current VLMs regarding their reliance on temporal priors, which leads to "chronological bias" and spatiotemporal hallucinations in dynamic environments. By identifying that embodied reasoning must evolve from basic spatial understanding to complex long-horizon planning, the authors provide a well-grounded motivation for developing a curriculum-based approach to task-oriented reasoning.
2. The analysis is extensive, supported by a large-scale dataset of 46 million samples and a Dual-Level Evaluation Framework that effectively decouples atomic perception from logical planning consistency. The experimental validation is comprehensive, featuring detailed ablation studies on the curriculum stages and the subtask planner.
3. The presentation is highly organized and clear, with intuitive figures that provide a high-level overview of the EgoTSR framework and its curriculum learning stages.

Weaknesses:
1. Motivation Unclear: The paper identifies "chronological bias" as a primary reason for spatiotemporal hallucinations in vision-language models. However, the link between this bias and the proposed three-stage curriculum learning paradigm lacks a rigorous theoretical foundation. While the authors present the "explicit-to-internalized" evolution as an imitation of human cognitive development, they do not sufficiently justify why this specific multi-stage strategy is essential or superior to simpler joint-training methods for correcting temporal priors at the algorithmic level.
2. Insufficient Comparison: The experimental evaluation is predominantly conducted on the self-constructed EgoTSR-Data and its corresponding Dual-Level Evaluation Framework. While the results show improvements over several baselines, the lack of validation on established, third-party benchmark datasets makes it difficult to assess the cross-dataset generalizability of the proposed framework. I would expect to see performance comparisons on at least one or two widely recognized ego-centric reasoning benchmarks to further substantiate the claims.
3. Complexity: The proposed EgoTSR framework involves a highly complex pipeline consisting of three distinct data types (CoT, Tag, LongTag), a three-stage sequential optimization process, and an auxiliary Reasoning-Enhanced Task Decomposition mechanism. This significant complexity, compared to end-to-end baselines, makes it challenging to decouple the contributions of the individual components. Specifically, it remains unclear whether the performance gains are derived from the structured curriculum architecture or simply the massive 46-million-sample scale of the training data.
4. Minor: The figures and diagrams provided in the manuscript appear to be rasterized images (JPG/PNG formats). Using vector graphics is standard practice to ensure clarity and professional presentation, especially for complex architectural overviews and data samples.

---

> ### Author Rebuttal · Authors · 2026-03-31
>
> ### **1. Why Curriculum Learning Can Correct Chronological Bias**
> We thank the reviewer for raising this insightful question. Our core motivation is that chronological bias is not solely manifested at the output layer, but is embedded within the internal representations formed during pre-training. Therefore, correcting this bias requires a decoupled, step-by-step reshaping of the model's intermediate representations. In contrast, simple joint training often mixes supervision signals across different temporal scales, leading to gradient interference and encouraging the model to learn shortcuts (e.g., relying on co-occurrence statistics) rather than genuine temporal reasoning capabilities. Without intermediate structural constraints, it is difficult for the model to learn decoupled representations of temporal dependencies.
>
> ---
> ### **2. Supplement to Recognized Ego-Centric Reasoning Benchmark Evaluation**
> **(1) Cross-Embodiment Generalization (OOD):** To explicitly test Out-of-Distribution (OOD) generalization, we evaluated our model—which was trained **exclusively on robotic arm data**—on **human-hands** datasets (EPIC-Kitchens [1] and Ego4D [2]). For brevity, we report key metrics: average accuracy (Avg), Confidence Interval (CI), standard deviation (Std), and significance testing (ST):
> |Model|E4D-S|E4D-L|EK-S|EK-L|Avg|CI|Std|ST|
> |-|:-:|:-:|:-:|:-:|:-:|:-:|:-:|:-:|
> |**Close-Source**|||||||||
> |GPT-5.4|50.5|63.0|56.5|66.9|59.2|[53.7,64.7]|0.097|0.23|
> |Gemini-3.1-flash|51.2|61.8|56.9|67.9|59.5|[54.9,64.1]|0.080|0.88|
> |Grok-4.1-fast|49.5|51.0|46.9|51.4|49.7|[46.7,52.7]|0.053|0.50|
> |**Open-Source-2D**|||||||||
> |NVILA-8b|50.2|48.7|49.5|48.5|49.2|[48.8,50.1]|0.499|0.02|
> |Qwen2.5-VL-7b|50.3|50.2|50.3|46.1|49.2|[47.7,48.8]|0.499|0.22|
> |LLaVA-OneVision|52.2|50.6|49.5|49.7|50.5|[48.1,52.3]|0.007|0.62|
> |**Open-Source-3D**|||||||||
> |3D-LLM|47.6|49.9|53.4|49.9|50.2|[48.9,51.5]|0.023|0.68|
> |LL3DA|49.1|49.8|50.7|49.3|49.7|[49.0,50.4]|1.260|0.65|
> |**Ours**|||||||||
> |**EgoTSR**|**56.0**|**59.0**|**54.0**|**57.0**|**57.5**|**[55.3,59.6]**|**0.494**|**---**|
>
> Results show EgoTSR outperforms other open-source model on EPIC-Kitchens [1], and Ego4D [2], strongly demonstrating its cross-dataset generalization and stability.
>
> **(2) Distinct Advantages on Fine-Grained Benchmarks:**
> We also evaluated on VSI-Bench [3], a third-party recognized benchmark, and the results strongly demonstrate the effectiveness of our method.
> |Models|Avg.|Obj. Count|Abs. Dist.|Obj. Size|Rel. Dist.|Rel. Dir.|Appr. Order|
> |-|:-:|:-:|:-:|:-:|:-:|:-:|:-:|
> |LongVILA-8B|21.6|29.1|9.1|16.7|29.6|30.7|25.5|
> |InternVL2-2B|26.5|25.7|24.0|20.0|32.1|44.1|6.3|
> |VILA-1.5-8B|28.9|17.4|21.8|50.3|32.1|34.8|24.8|
> |**EgoTSR**|**24.8**|**24.1**|**21.9**|**21.9**|**47.1**|**31.4**|**29.4**|
>
> ---
> ### **3. Curriculum Learning Necessity**
> We compare the base model (before training), unordered mixed training, and our curriculum approach:
> |Model|Forward|Inverse|Avg|Gap|
> |-|:-:|:-:|:-:|:-:|
> |Base Model|55.6|49.8|52.7|5.8|
> |Unordered Mixed Training|67.2|72.1|69.6|4.9|
> |**Curriculum Training**|**92.4**|**92.3**|**92.4**|**0.1**|
>
> Results show that the curriculum sequence—not mere data scaling—is crucial for the significant performance boost.
>
> ---
>
> ### **4. Image Format Issue**
> We have a large number of complex figures. Exporting all of them as vector graphics would exceed the submission file size limit. We will do our best to utilize vector graphics in the camera-ready version.
>
> ---
>
> ### **5. How Does Our Method Embody the "Ego-Centric" Perspective?**
> Our method reflects the "ego-centric" perspective primarily through data sources, input representations, and the core design of the evaluation.
>
> **(1) Data Construction Based on a First-Person Perspective:** The data for our foundational dataset (EgoTSR-Data) originates from the agibot-world dataset, which provides a massive amount of high-fidelity, large-scale "ego-centric" data of real-world robotic manipulation.
>
> **(2) Ego-Centric Model Input Perspective:** The visual observation input for the EgoTSR model is explicitly defined as two frames from an "ego-centric" video sequence. Furthermore, the task names in the text prompts, the spatial perception relationships between the robotic arm and the surrounding scene during the CoT (Chain-of-Thought) stage, and the subtask sequence descriptions during the long-tag stage are all formulated from an ego-centric viewpoint.
>
> **(3) Benchmark Evaluation Targeting the First-Person Perspective:** One of the main objectives of the Dual-Level Evaluation Framework proposed in the paper is to verify whether the model can strike a balance between local spatial perception and global temporal planning when processing "first-person perspective tasks."
>
> ---
>
> ### **References**
> [1] Scaling egocentric vision: The epic-kitchens dataset.
>
> [2] Ego4d: Around the world in 3,000 hours of egocentric video.
>
> [3] Thinking in space: How multimodal large language models see, remember, and recall spaces.

---

> > ### Author Rebuttal · Reviewer_cjEN · 2026-04-01
> >
> > 1. Figures in the paper should be in **.pdf format**, which is standard and common in conference papers. Figs. 2, 3, 5, and 6 do not contain many elements, so it is easy for authors to offer the .pdf format. Figs.1 and 4 indeed contain many images, but figures in the pdf format can be compressed with keeping enough legibility. Many open websites or tools can help it. It is not professional if the authors keep figures in .jpg/.png format.
> > 2. The response to W1 offers an intuitive motivation, but not supported by sufficient evidences. This question may be hard to solve, and the ituitive motivation is enough for this paper under weak accept .
> >
> > I have revised the score, but also degraded the confidence. I will restore the confidence if authors are willing to offer .pdf figures in the camera-ready version for professionalism.
> >
> > ### Update 4.4
> > I have revised my confidence score. Thanks.

---

> > > ### Author Response · Authors · 2026-04-02
> > >
> > > First of all, we sincerely thank you for your rigorous attention to detail, constructive suggestions, and the in-depth discussion regarding the motivation of our paper.
> > >
> > > ---
> > >
> > > ### **1. Figure Formats**
> > >
> > > We appreciate your helpful suggestion regarding the figure format. We will ensure that all figures are provided in .pdf format in the camera-ready version.
> > >
> > > ---
> > >
> > > ### **2. Further Theoretical Analysis**
> > >
> > > To theoretically ground why progressive curriculum learning outperforms joint training in mitigating chronological bias, we detail our motivation across four dimensions:
> > >
> > > **(1) A Holistic Perspective on Curriculum Learning**
> > >
> > > Curriculum learning constructs a **smooth optimization path** from simple to complex objectives [1]. Let the target risk be: $\mathcal{L}(\theta)=\mathbb{E}_{z\sim P}[\ell(\theta;z)],$
> > >
> > > where $P$ is the target data distribution. Rather than directly optimizing $\mathcal{L}$, curriculum learning introduces reweighted training distributions: $Q_\lambda(z)\propto W_\lambda(z)P(z), \quad \lambda\in[0,1],$
> > >
> > > with the staged objective function: $\mathcal{L}_\lambda(\theta)=\mathbb{E}[\ell(θ;z)\mid z\sim Q(λ)].$
> > >
> > > As $\lambda$ increases, training transitions through: $\min_\theta \mathcal{L}(θ, λ_0)\rightarrow\min_\theta \mathcal{L}(θ, λ_1)\rightarrow\cdots\rightarrow\min_\theta \mathcal{L}(θ, 1).$
> > >
> > > This **continuation method** optimizes smoother objectives first, facilitating convergence into favorable basins during non-convex optimization.
> > >
> > > **(2) Reducing Gradient Noise and Conflicts**
> > >
> > > Curriculum learning also mitigates Stochastic Gradient Descent (SGD) noise [2]. Let the parameter update be: $\theta_{t+1}=\theta_t-\eta g_t, \quad \mathbb{E}[g_t]=\nabla \mathcal{L}_\lambda(\theta_t),$
> > >
> > > with gradient noise covariance: $\Sigma_\lambda(\theta)=\mathrm{Var}\big[\nabla_\theta \ell(\theta;z)\mid z\sim Q(λ)\big].$
> > >
> > > Introducing difficult samples early causes higher gradient variance and stronger conflicting directions. Under local strong convexity, the SGD error bound is: $\mathbb{E}\|\theta_T-\theta^\star\|^2\lesssim(1-\eta\mu)^T\|\theta_0-\theta^\star\|^2+\frac{\eta}{\mu}\mathrm{tr}(\Sigma).$
> > >
> > > By **delaying highly difficult samples**, curriculum learning curtails early gradient noise, enhancing optimization stability.
> > >
> > > **(3) Staged Warm-Start Mechanism**
> > >
> > > Curriculum learning provides a **staged warm-start mechanism** where preceding parameters initialize the next stage. Let the optimal solution at stage $k$ be $\theta_k^\star$. The warm-start initialization is: $\theta_{k}^{(0)} = \theta_{k-1}^\star.$
> > >
> > > Under local strong convexity, the optimization error satisfies: $\|\theta_t - \theta_k^\star\|^2\le (1-\eta\mu)^t \|\theta_{k}^{(0)} - \theta_k^\star\|^2.$
> > >
> > > Compared to random initialization $\theta^{(0)} \sim \mathcal{D}$, we have: $\|\theta_{k}^{(0)} - \theta_k^\star\|\ll\|\theta^{(0)} - \theta_k^\star\|.$
> > >
> > > This minimizes initial error, accelerates convergence, and aligns gradient directions. Introducing complex tasks only after establishing fundamental visual logic further boosts performance.
> > >
> > > **(4) Why Curriculum Learning Mitigates Chronological Bias**
> > >
> > > Joint training with temporally balanced data constrains the final objective $\mathcal{L}(\theta)$, but not the optimization trajectory. SGD preferentially converges toward low-complexity shortcut solutions with stable gradients, rather than genuine spatiotemporal reasoning [1,3]. Thus, even with balanced data, a jointly trained model may still rely on chronological shortcuts.
> > >
> > > **Curriculum learning explicitly guides the optimization trajectory to prevent this:**
> > > * **Early stage:** Reinforces visual evidence via explicit CoT constraints (avoiding shortcuts).
> > > * **Middle stage:** Internalizes reasoning capabilities via intuitive perception.
> > > * **Late stage:** Introduces complex temporal relationships to solidify long-term reasoning.
> > >
> > > This staged approach prevents premature convergence to shortcuts. Conceptually: $\text{Curriculum}\approx\text{staged continuation}+\text{warm start}+\text{gradient-noise control}.$
> > >
> > > By stabilizing optimization, reducing noise, and suppressing shortcuts, these mechanisms fundamentally alleviate **chronological bias**.
> > >
> > > ---
> > >
> > > We hope that our response helps clarify your concerns and meets your expectations.
> > >
> > > ---
> > >
> > > **References**
> > >
> > > [1] Y. Bengio et al., *Curriculum Learning*. ICML, 2009.
> > > [2] Z. Xu and A. Tewari, *On the Statistical Benefits of Curriculum Learning*. ICML, 2022.
> > > [3] R. Geirhos et al., *Shortcut Learning in Deep Neural Networks*. Nature Machine Intelligence, 2020.

---

### Official Review · Reviewer_4CKB · 2026-03-18

**Soundness:** 3
**Presentation:** 2
**Significance:** 3
**Originality:** 2
**Overall Recommendation:** 4
**Confidence:** 3

**Summary:**

This paper addresses the limitations of spatiotemporal hallucination, chronological bias and poor long‑horizon planning in egocentric task‑oriented reasoning for embodied AI. It proposes the EgoTSR framework, which adopts a three‑stage curriculum learning paradigm built on a 46‑million‑sample EgoTSR‑Data dataset. The authors design a reasoning‑enhanced task decomposition mechanism to map abstract tasks to atomic subtasks, and establish a dual‑level evaluation framework for spatial perception and logical inference. Experiments show EgoTSR effectively mitigates chronological bias, achieves superior performance on long‑horizon reasoning and fine‑grained perception, and is validated on simulation and real‑robot platforms.

**Compliance With Llm Reviewing Policy:**

Affirmed.

**Key Questions For Authors:**

1. Is the sample ratio and training sequence of the three-stage curriculum learning adjustable? What is its impact on model performance?
2. Can the reasoning-enhanced task decomposition mechanism maintain effectiveness for complex cross-scene tasks?
3. How to further improve the model robustness under severe occlusion or dynamic interference in real open scenes?
4. What is the inference latency and computational overhead of EgoTSR, and is it friendly for real-time robot deployment?

**Limitations:**

1. The model is fine-tuned based on existing vision-language models without novel architectural innovations, and core contributions focus on training paradigms and dataset construction.
2. The proportion and parameter settings of the three-stage curriculum learning lack systematic ablation analysis, and its adjustability and boundary conditions are not fully discussed.
3. Validation is only conducted on a small number of robot platforms, lacking robustness verification under complex dynamic open scenes and severe occlusion.
4. The inference latency and computational overhead of the model are not analyzed, so the efficiency adaptability for real-time robot deployment is not demonstrated.

**Strengths And Weaknesses:**

Strengths
+ Aiming at spatiotemporal hallucination and chronological bias in embodied AI, this paper proposes a three-stage curriculum learning paradigm that conforms to human cognitive evolution with a reasonable design.
+ It builds the EgoTSR dataset with 46 million samples, which is staged to meet training demands and provides solid data support.
+ The reasoning-enhanced task decomposition mechanism effectively bridges high-level task semantics and low-level robot execution, and the dual-level evaluation framework is rigorous.
+ Comprehensive experiments are conducted against mainstream closed-source and open-source models, with effectiveness verified on simulation and real-robot platforms and significant performance gains.

Weaknesses
- The model is fine-tuned based on Qwen-VL-7B with limited architectural innovation, and core contributions focus on data and training paradigms.
- Long-horizon generalization is only validated on a few robot platforms, lacking robustness analysis in complex dynamic open scenes.
- Ablation studies of curriculum learning stages are not in-depth, and discussions on training parameters and boundary conditions are brief.

---

> ### Author Rebuttal · Authors · 2026-03-31
>
> ### **1. Curriculum Learning Sample Ratio and Necessity Analysis**
> **(1) Sample Ratio:** We conducted an ablation analysis on the sample ratio of CoT data to Tag data used for training, and evaluated the accuracy for short and long tasks. The results are as follows:
>
> | CoT : Tag (%) | 14.3 | 28.6 | 42.9 | 57.1 | 71.4 | 85.7 | 100.0 | 114.3 | 128.6 | 142.9 | 157.1 | 171.4 |
> | :--- | :-: | :-: | :-: | :-: | :-: | :-: | :-: | :-: | :-: | :-: | :-: | :-: |
> | **Accuracy (Short)** | 54.9 | 62.0 | 73.5 | 74.9 | 76.5 | 77.8 | 80.0 | 80.7 | 81.8 | 83.0 | 82.8 | 84.7 |
> | **Accuracy (Long)** | 57.5 | 67.1 | 78.8 | 82.5 | 84.8 | 85.0 | 85.3 | 85.8 | 87.0 | 86.8 | 87.7 | 88.9 |
> | **Overall** | 56.2 | 64.5 | 76.1 | 78.7 | 80.6 | 81.4 | 82.6 | 83.2 | 84.4 | 84.9 | 85.2 | 86.8 |
>
> **(2) Curriculum Necessity:** Regarding the curriculum learning sequence, we evaluated three models: the base model prior to fine-tuning, a model trained with unordered mixed sampling without stages, and a model trained following the curriculum learning sequence. The experimental results are as follows:
>
> |Model|Forward|Inverse|Avg|Gap|
> |:---|:---:|:---:|:---:|:---:|
> |Baseline|55.6|49.8|52.7|5.8|
> |Unordered Training|67.2|72.1|69.6|4.9|
> |**Curriculum Training (Ours)**|**92.4**|**92.3**|**92.4**|**0.1**|
>
> The results show that **curriculum learning outperforms simple joint training methods**, rather than just relying on data scaling.
>
> ---
>
> ### **2. Experimental Validation of the Reasoning-Enhanced Task Decomposition  Mechanism Maintaining Effectiveness for Complex Cross-Scene Tasks**
> We evaluated it using VLABench [1] across Mesh & Texture (M&T), Semantic Understanding (Sem), Spatial Awareness (Spa), Physics Law (Phy), Complex Reasoning (Cpx), and Commonsense (CS). The experimental results are as follows:
>
> | Model | Average | M&T | Sem | Spa | Phy | Cpx | CS |
> | :--- | :---: | :---: | :---: | :---: | :---: | :---: | :---: |
> | Qwen3-VL (baseline) | 25.10 | 29.6 | 25.9 | 31.1 | 24.0 | 14.3 | 25.7 |
> | MiniCPM-V-2.6 | 25.96 | 31.0 | 24.1 | 30.6 | 18.3 | 21.8 | 30.0 |
> | Qwen2.5-VL-3B | 23.18 | 27.1 | 22.2 | 32.5 | 17.4 | 13.9 | 25.9 |
> | InternVL2.5-2B | 20.24 | 28.4 | 24.2 | 24.1 | 4.5 | 14.5 | 25.6 |
> | Qwen2-VL-2B | 17.37 | 23.0 | 22.2 | 23.8 | 2.4 | 11.3 | 21.5 |
> | **Subtask Planner (Ours)** | **28.38** | **32.8** | **28.5** | **31.6** | **30.7** | **20.2** | **27.3** |
>
> ---
>
> ### **3. Discussion on "How to further improve the model's robustness under severe occlusion or dynamic interference in real-world open scenes"**
> We thank the reviewer for raising this insightful question. EgoTSR aggregates temporal information, which effectively compensates for missing observations. To further improve robustness in real-world open scenes, we can explore the following directions:
>
> **(1)** Integrating multimodal or multi-view information to reduce dependence on a single visual input.
>
> **(2)** Enhancing data augmentation by simulating occlusions and interferences to improve the model's robustness.
>
> ---
> ### **4. Real-Time Robot Deployment and Computational Overhead**
> EgoTSR is highly suitable for real-robot deployment. Specifically, a single inference takes only about 1 second to process long-horizon visual inputs—ranging from dozens of frames for short tasks to hundreds of frames for long tasks. Regarding computational overhead, inference requires only a single consumer-grade RTX 4090 GPU, with peak VRAM usage strictly under 20GB. This high processing efficiency, combined with lightweight hardware requirements, offers a highly cost-effective solution for real-world applications.
>
> ---
>
> ### **References**
> [1] Vlabench: A large-scale benchmark for language-conditioned robotics manipulation with long-horizon reasoning tasks.

---

> > ### Author Rebuttal · Reviewer_4CKB · 2026-04-05
> >
> > My concerns have been adequately addressed.

---

> > > ### Author Response · Authors · 2026-04-05
> > >
> > > We are very pleased that your concerns have been adequately addressed. We sincerely thank you for the valuable time and effort you devoted to reviewing our work. Should you have any further questions, we would greatly appreciate your continued feedback.

---

### Decision · Program_Chairs · 2026-04-30

**Decision:**

Accept (regular)

**Comment:**

The paper presents a curriculum-based framework for ego-centric task-oriented spatiotemporal reasoning demonstrating empirical gains across multiple datasets and evaluation protocols. The reviews and discussion indicate a clear positive consensus, with all reviewers converging to weak accept after the rebuttal and explicitly stating that their main concerns were fully resolved.  Remaining issues are acknowledged but do not undermine the technical soundness or the utility of the dataset and evaluation framework for the community.  Given the consensus, I recommend acceptance.